# SIMULTANEOUSLY LEARNING STOCHASTIC AND AD­VERSARIAL MARKOV DECISION PROCESS WITH LIN­EAR FUNCTION APPROXIMATION

## ABSTRACT

Reinforcement learning (RL) has been commonly used in practice. To deal with the numerous states and actions in real applications, the function approximation method has been widely employed to improve the learning efficiency, among which the linear function approximation has attracted great interest both theoretically and empirically. Previous works on the linear Markov Decision Process (MDP) mainly study two settings, the stochastic setting where the reward is generated in a stochastic way and the adversarial setting where the reward can be chosen arbitrarily by an adversary. All these works treat these two environments separately. However, the learning agents often have no idea of how rewards are generated and a wrong reward type can severely disrupt the performance of those specially designed algorithms. So a natural question is whether an algorithm can be derived that can efficiently learn in both environments but without knowing the reward type. In this paper, we first consider such best-of-both-worlds problem for linear MDP with the known transition. We propose an algorithm and prove it can simultaneously achieve $O(\text{poly} \log K)$ regret in the stochastic setting and $\tilde{O}(\sqrt{K})$ regret in the adversarial setting where $K$ is the horizon. To the best of our knowledge, it is the first such result for linear MDP.

## 1 INTRODUCTION

Reinforcement learning (RL) studies the problem where a learning agent interacts with the environ­ment over time and aims to maximize its cumulative rewards in a given horizon. It has a wide range of real applications including robotics (Kober et al., 2013), games (Mnih et al., 2013; Silver et al., 2016), etc. The environment dynamics are usually modeled by the Markov Decision Process (MDP) with a fixed transition function. We consider the general episodic MDP setting where the interactions last for several episodes and the length of each episode is fixed (Jin et al., 2018; 2020b; Luo et al., 2021; Yang et al., 2021). In each episode, the agent first observes its current state and would decide which action to take. After making the decision, it receives an instant reward and the environment will then transfer to the next state. The cumulative reward in an episode is called the value and the objective of the agent is equivalent to minimizing the regret defined as the cumulative difference between the optimal value and its received values over episodes.

Many previous works focus on the tabular MDP setting where the state and action space are finite and the values can be represented by a table (Azar et al., 2017; Jin et al., 2018; Chen et al., 2021; Luo et al., 2021). Most of them study the stochastic setting with the stationary reward in which the reward of a state-action pair is generated from a fixed distribution (Azar et al., 2017; Jin et al., 2018; Simchowitz & Jamieson, 2019; Yang et al., 2021). Since the reward may change over time in applications, some works consider the adversarial MDP where the reward can be arbitrarily generated among different episodes (Yu et al., 2009; Rosenberg & Mansour, 2019; Jin et al., 2020a; Chen et al., 2021; Luo et al., 2021). All of these works pay efforts to learn the value function table to find the optimal policy and the computation complexity highly depends on the state and action space size.

However, in real applications such as the Go game, there are numerous states and the value function table is huge, which brings a great challenge to the computation complexity for traditional algorithms in the tabular case. To cope with the dimensionality curse, a rich line of works employ the function

approximation methods, such as the linear function and deep neural networks, to approximate the value functions or the policies to improve learning efficiency. These methods also achieve great success in practical scenarios such as the Atari and Go games (Mnih et al., 2013; Silver et al., 2016). Despite their great empirical performances, it also brings a series of challenges in deriving theoretical analysis. To build a better theoretical understanding of these approximation methods, lots of works start from deriving regret guarantees for linear function classes.

The linear MDP is a popular model which assumes both the transition and reward at a state-action pair are linear in the corresponding $d$-dimensional feature (Jin et al., 2020b; He et al., 2021; Hu et al., 2022). There are also mainly two types of the reward. For the stochastic setting, Jin et al. (2020b) provides the first efficient algorithm named Least-Square Value Iteration UCB (LSVI-UCB) and show that the its regret over $K$ episodes can be upper bounded by $O(\sqrt{K})$. To seek for a tighter result with respect to the specific problem structure, He et al. (2021) provide a new analysis for LSVI-UCB and show it achieves an $O(\text{poly} \log K)$ instance-dependent regret upper bound. The adversarial setting is much harder than the stochastic one since the reward can change arbitrarily but the agent can only observe the rewards on the experienced trajectory. For this more challenging case, a regret upper bound of order $O(\sqrt{K})$ is only obtained in the case with known transition by Neu & Olkhovskaya (2021). All these works separately treat two environment types.

However, the learning agent usually has no idea of how the reward is generated. And once the reward type is wrong, the specially designed algorithm for a separate setting may suffer great loss. Thus deriving an algorithm that can adapt to different environment types becomes a natural solution for this problem. This direction has attracted great research interest in simpler bandit (Bubeck & Slivkins, 2012; Zimmert et al., 2019; Lee et al., 2021; Kong et al., 2022) and tabular MDP settings (Jin & Luo, 2020; Jin et al., 2021b) but still remains open in linear MDP.

In this paper, we try to answer the question of deriving best-of-both-worlds (BoBW) guarantees for linear MDP. Due to the challenge of learning in the adversarial setting, we also consider the known transition case. We propose an algorithm that continuously detects the real environment type and adjusts its strategy. It has been shown that our algorithm can simultaneously achieve $O(\text{poly} \log K)$ regret in the stochastic setting and $\tilde{O}(\sqrt{K})$ regret in the adversarial setting. To the best of our knowledge, these are the first BoBW results for linear MDP. It is also worth noting that our BoBW algorithm relies on an algorithm that can achieve a high-probability guarantee for the adversarial setting, which previous works fail to provide. And we propose the first analysis for a high-probability regret bound in the adversarial linear MDP.

## 2 RELATED WORK

**Linear MDP.** Recently, deriving theoretically guaranteed algorithms for RL with linear function approximation has attracted great interests. The linear MDP model is one of the most popular one. Jin et al. (2020b) develop the first efficient algorithm LSVI-UCB both in sample and computation complexity for this setting. They show that the algorithm achieves $O(\sqrt{d^3 H^3 K})$ regret where $d$ is the feature dimension and $H$ is the length of each episode. This result is recently improved to the optimal order $O(dH\sqrt{K})$ by Hu et al. (2022) with a tighter concentration analysis. Apart from UCB, the TS-type algorithm has also been proposed for this setting (Zanette et al., 2020a). All these results do not consider the specific problem structure. In the stochastic setting, deriving an instance-dependent regret is more attractive to show the tighter performances of algorithms in a specific problem. This type of regret has been widely studied under the tabular MDP setting (Simchowitz & Jamieson, 2019; Yang et al., 2021). He et al. (2021) is the first to provide this type of regret in linear MDP. Using a different proof framework, they show that the LSVI-UCB algorithm can achieve $O(d^3 H^5 \log K/\Delta)$ where $\Delta$ is the minimum value gap in the episodic MDP.

All these works consider the stochastic setting with stationary rewards. Neu & Olkhovskaya (2021) first attempts to analyze the more challenging adversarial environment. They consider a simplier setting with known transition and provide an $O(\sqrt{dHK})$ regret upper bound. For unknown transition case, Luo et al. (2021) provide an $O(d^{2/3}H^2 K^{2/3})$ upper bound with the help of a simulator and $O(d^2 H^4 K^{14/15})$ guarantee for the general case. Above all, even in the separate adversarial setting, $O(\sqrt{K})$ regret is only derived for known transition case. We also study the known transition

setting and try to provide $\tilde{O}(\sqrt{K})$ regret in the adversarial setting while simultaneously achieving $O(\text{poly} \log K)$ regret if the environment is truly stochastic.

**Best-of-both-worlds.** The question of reaching best-of-both-worlds results is first proposed by Bubeck & Slivkins (2012) for bandit setting, a special case of episodic MDP with $H = 1$. Their proposed algorithm assumes the setting is stochastic and continuously detects whether the assumption is satisfied. Such a detection-based method is shown to achieve $O(\text{poly} \log K)$ regret in the stochastic setting and $O(\sqrt{K})$ in the adversarial setting, which is later improved by Auer & Chiang (2016). Similar detection-based techniques have also been adopted in more general linear bandit (Lee et al., 2021) and graph feedback (Kong et al., 2022) settings to achieve BoBW guarantees. Another line of works consider using Follow-the-Regularized-Leader (FTRL) to adapt to different environment types (Zimmert & Seldin, 2019; 2021). This type of algorithm is shown to be tighter than Bubeck & Slivkins (2012); Auer & Chiang (2016) in the bandit setting and also attracts lots of interest in more complex problems such as combinatorial bandits (Zimmert et al., 2019; Chen et al., 2022).

The first BoBW result in the MDP setting is provided by Jin & Luo (2020) in the tabular case. Due to the challenge of the problem, they first study the known transition setting. Their approach to achieving BoBW is the FTRL algorithm with a newly designed regularizer, which result is later improved by Jin et al. (2021b) and also generalized to the unknown transition case. To the best of our knowledge, we are the first to consider the BoBW problem in linear MDP. We also start from the known transition setting and our algorithm is based on detection.

**RL with general function approximation** The linear mixture MDP is another popular RL model with linear function approximation. It assumes the transition function can be approximated by a weighted average over several transition kernels. In the stochastic setting, both instance-independent (Ayoub et al., 2020; Zhou et al., 2021) and dependent regret bound (He et al., 2021) have been derived. And in the adversarial setting, only full information case has been studied, where the agent has access to the rewards of all state-action pairs (Cai et al., 2020; He et al., 2022). Apart from linear function approximation, there is also a rich line of works considering general function classes, such as the setting with low Bellman rank (Jiang et al., 2017; Zanette et al., 2020b), low Eluder dimension (Wang et al., 2020; Kong et al., 2021) and low Bellman Eluder dimension (Jin et al., 2021a).

## 3 SETTING

We consider the episodic MDP setting where the agent interacts with the environment for $K$ episodes with known transition. The episodic MDP can be represented by $\mathcal{M}(\mathcal{S}, \mathcal{A}, H, \{r_k\}_{k=1}^K, P)$ where $\mathcal{S}$ is the state space, $\mathcal{A}$ is the action space, $H$ is the length of each episode, $r_k = \{r_{k,h}\}_{h=1}^H$ is the reward function and $P = \{P_h\}_{h=1}^H$ is the known transition probability function. Specifically, at each episode $k$ and step $h \in [H]$, $r_{k,h}(s,a) \in [0,1]$ and $P_h(\cdot \mid s,a) \in [0,1]^{|\mathcal{S}|}$ are the reward and transition probability at state $s \in \mathcal{S}$ by taking action $a \in \mathcal{A}$, respectively.

We focus on stationary policies. Denote $\pi = \{\pi_h\}_{h=1}^H$ as a policy mapping from the state space to an action distribution where $\pi_h : \mathcal{S} \to \Delta_{\mathcal{A}}$. For each episode $k \in [K]$, the agent would start from the first state $s_{k,1} := s_1$[1] and determine the policy $\pi_k$. Then at each step $h \in [H]$ of episode $k$, it first observes the current state $s_{k,h}$ and then perform the action $a_{k,h} \sim \pi_{k,h}(\cdot \mid s_{k,h})$. The agent would receive a random reward $y_{k,h} := r_{k,h}(s_{k,h}, a_{k,h}) + \epsilon_{k,h}$, where $\epsilon_{k,h}$ is an independent zero-mean noise. The environment then transfers to the next state $s_{k,h+1}$ based on the transition function $P(\cdot \mid s_{k,h}, a_{k,h})$. The episode ends when the last state $s_{k,H+1}$ is reached.

We focus on linear MDP with known transition where the reward functions are linear in a given feature mapping (Jin et al., 2020b; He et al., 2021). The formal definition is as follows.

**Assumption 1.** *(Linear MDP with known transition) $\mathcal{M}(\mathcal{S}, \mathcal{A}, H, \{r_k\}_{k=1}^K, P)$ is a linear MDP with a known feature mapping $\phi : \mathcal{S} \times \mathcal{A} \to \mathbb{R}^d$ such that for each step $h \in [H]$, there exists an unknown vector $\theta_h$ where for each $(s,a) \in \mathcal{S} \times \mathcal{A}$, $r_{k,h}(s,a) = \langle \phi(s,a), \theta_{k,h} \rangle$.*

---

[1]The deterministic starting state is only for expositional convenience. Our algorithm and analysis can directly handle random starting states with a distribution.

In the stochastic setting, the reward function $\{r_k\}_{k=1}^K$, or namely the reward parameter $\{\theta_k\}_{k=1}^K$, is fixed over different episodes $k \in [K]$. And in the adversarial setting, the reward parameter $\{\theta_k\}_{k=1}^K$ can be chosen arbitrarily by an adversary (which may be possibly dependent on previous policies).

We evaluate the performance of a policy $\pi$ by its value functions. Specifically, for any episode $k$ and step $h$, denote the $Q$-value function $Q_{k,h}^\pi(s, a)$ as the expected reward that will be obtained by the agent starting from $(s_{k,h}, a_{k,h}) = (s, a)$ with policy $\pi$, which is formally defined as

$$Q_{k,h}^\pi(s, a) = \mathbb{E}\left[\sum_{h'=h}^H y_{k,h'} \mid \pi, s_{k,h} = s, a_{k,h} = a\right].$$

Similarly, the value function $V_{k,h}^\pi(s)$ of any state $s$ is defined as

$$V_{k,h}^\pi(s) = \mathbb{E}\left[\sum_{h'=h}^H y_{k,h'} \mid \pi, s_{k,h} = s\right].$$

In the following paper, we abuse a bit notation by using $V_k^\pi := V_{k,1}^\pi(s_1)$ to represent the value of policy $\pi$ at the starting state $s_1$ and episode $k$. Define $\phi_{\pi,h} = \mathbb{E}\left[\phi(s_h, a_h) \mid \pi\right]$ as the expected feature vector that the policy $\pi$ visits at step $h$. It is worth noting that this recovers the state-action visitation probability in the tabular setting. And according to the definition, the value function of policy $\pi$ at episode $k$ can be represented as $V_k^\pi = \sum_{h=1}^H \langle \phi_{\pi,h}, \theta_{k,h} \rangle$.

In this paper, we consider optimizing both the stochastic and adversarial environments within a finite policy set $\Pi$. Given a policy set $\Pi$, the learning agent would determine the policy $\pi_k \in \Pi$ in each episode $k \in [K]$. Let $\pi^* \in \arg\max_\pi \sum_{k=1}^K V_k^\pi$ be one optimal policy in $\Pi$ that maximizes the cumulative value functions over $K$ episodes, which is assumed to be unique in the stochastic setting similar to previous works in tabular MDP (Jin & Luo, 2020; Jin et al., 2021b) and bandit setting (Lee et al., 2021; Zimmert & Seldin, 2019; 2021). Denote the cumulative regret compared with $\pi^* \in \Pi$ over $K$ episodes as

$$Reg(K; \Pi) = \sum_{k=1}^K \left(V_k^{\pi^*} - V_k^{\pi_k}\right). \tag{1}$$

## 4 ALGORITHM

In this section, we propose a detection-based algorithm to optimize both stochastic and adversarial environments for linear MDP with a given policy set $\Pi$. Our algorithm is mainly inspired by the detection technique of Lee et al. (2021) for BoBW in linear bandits. At a high level, the algorithm first assumes the environment is truly adversarial and continuously detect whether it could be a stochastic one. Its design relies on a new linear MDP algorithm that can return well-concentrated estimators for values of policies and also achieve sub-linear regret in the adversarial setting with high probability.

Previous works on adversarial linear MDP fail to provide a high-probability guarantee and thus no existing algorithms satisfy this property. In Appendix D, we propose a variant of Geometric Hedge (Algorithm 4), which is initially designed for the simple bandit case (Bartlett et al., 2008), and provide a new analysis for it in the linear MDP setting. We show that this algorithm satisfies the following properties and can be used to derive the BoBW results. It is also worth noting that this algorithm is the first to achieve a high-probability regret guarantee for adversarial linear MDP.

**Theorem 1.** *Given a policy set $\Pi$, the regret of our proposed Algorithm 4 in the adversarial setting can be upper bounded by*

$$\mathrm{Reg}(K; \Pi) \leq \mathcal{O}\left(\sqrt{dH^3 K \log\left(|\Pi|/\delta\right)}\right) \tag{2}$$

*with probability at least $1 - \delta$.*

*Further, at each episode $k$, Algorithm 4 returns a value estimator $\hat{V}_k^\pi$ for each policy $\pi \in \Pi$. Choosing constant $L_0 = 4dH \log\left(|\Pi|/\delta\right)$, $C_1 \geq 2^{15} dH^3 \log\left(K|\Pi|/\delta\right)$ and $C_2 \geq 20$, it holds that for any*

$k_0 \geq L_0$ and policy $\pi \in \Pi$,

$$\sum_{k=1}^{k_0} \left(V_k^\pi - V_k^{\pi_k}\right) \leq \sqrt{C_1 k_0} - C_2 \left|\sum_{k=1}^{k_0} \left(\hat{V}_k^\pi - V_k^\pi\right)\right|$$

with probability larger than $1 - \delta$.

Our main BoBW algorithm is a phased algorithm and is presented in Algorithm 1. It takes Algorithm 4 satisfying Theorem 1 with parameter $L_0$ and $C_1$ as input. The first epoch is of length $L_0$ and the length of the following epochs would grow exponentially as Line 4. During each epoch, it executes Algorithm 2, which we refer to as the BoBW main body (Line 3).

---

**Algorithm 1** BoBW for linear MDP

1: Input: Algorithm 4 with parameter $C_1$ and $L_0$; Set $L := L_0$. Maximum duration $K$.
2: **while** number of episodes $k \leq K$ **do**
3:     Execute Algorithm 2 (BoBW main body) with parameter $L$ and receive output $k_0$
4:     Set $L = 2k_0$
5: **end while**

---

Algorithm 2 (BoBW main body) takes the Algorithm 4 with parameter $C_1$ and $L$ as input. Here $L$ can just be regarded as the minimum number of episodes that Algorithm 4 needs to run to collect enough observations. The algorithm first assumes the environment is adversarial and executes Algorithm 4 in at least $L$ episodes (which we refer to as the first phase). As shown in Theorem 1, Algorithm 4 guarantees that when running for more than $L$ episodes, the regret compared with any policy $\pi$ and the distance between its estimated $V$ value and the real $V$ value would be no larger. Based on these concentration properties, if a policy $\hat{\pi}$ shows consistent better performance than all of the other policies (Line 5), we have the reason to believe that the environment is truly stochastic. Being aware of this, as shown in Line 6, Algorithm 2 would transfer to the stochastic phase (Line 9-18, which we refer to as the second phase) with the estimated $V$ values returned by Algorithm 4.

Since the estimated values by Algorithm 4 can well approximate the real values of policies, the exploration in the stochastic setting can be conducted by the estimated value gaps to obtain a problem-dependent regret bound. The objective is to identify the optimal policy and maximize the collected rewards, which can be implemented by an optimization problem (Algorithm 3). Taking the estimated value gap $\hat{\Delta}$ as input, Algorithm 3 would return a probability distribution $p^*$ over the policy set $\Pi$. Intuitively, $p^*$ maximizes the expected values of policies while ensuring the uncertainty of all policies to be smaller than its sub-optimality gap. In Appendix A, we show that when $\hat{\Delta}$ is estimated accurately, selecting policies based on $p^*$ can reach a problem-dependent regret upper bound.

Back to the main body of Algorithm 2, after computing the distribution $p_k$ based on the current estimated $\hat{\Delta}$ (Line 10), it also mixes this policy distribution with a one-hot vector $e_{\hat{\pi}}$ to ensure that the estimated optimal policy $\hat{\pi}$ can be observed for enough times and the variance of its following estimators can thus be low (Line 11). The algorithm then samples a policy $\pi_k$ according to this mixed distribution and executes it in this episode. Then based on the received rewards $y_{k,h}$ at each step $h$ and the total reward $Y_k = \sum_{h=1}^{H} y_{k,h}$, the value estimations of policies can be further updated. Due to the technical reason, here for the estimated optimal policy $\hat{\pi}$ and other policies $\pi$, we use different estimators. Specifically, we use the importance weighted estimator to approximate the value of $\hat{\pi}$. The reason for using mixed policy distribution $\tilde{p}$ is just to ensure the low variance of this estimator. And for other policies, we use the standard least square estimators (Line 13). Based on these newly estimated values of policies, the algorithm can then update the estimation for their sub-optimality gaps as Line 14. To get a tighter analysis, when computing the value gap of $\pi$, we use the traditional estimator of it by Algorithm 4 in the first $k_0$ episodes and the Catoni estimator for the recent $k - k_0$ episodes. Formally speaking, the Catoni estimator **Catoni**$_\alpha \left(\{X_1, X_2, \cdots X_n\}\right)$ is defined as the unique root of $f(z) = \sum_{i=1}^{n} \Phi\left(\alpha\left(X_i - z\right)\right)$, where $\Phi(y) = \log\left(1 + y + y^2/2\right)$ if $y \geq 0$ and $\Phi(y) = -\log\left(1 - y + y^2/2\right)$ otherwise. The hyper-parameter $\alpha_k^\pi$ is set as $\sqrt{4 \log\left(k|\Pi|/\delta\right)/\sum_{\kappa=k_0+1}^{k} \left(2\kappa\hat{\Delta}_\pi^2/\beta_\kappa + 9dH\right)}$ which mainly follows Lee et al. (2021) but with the careful consideration in MDP setting.

---

**Algorithm 2** BOBW main body

1: Input: A new instance of Algorithm 4 with parameter $C_1$, parameter $L$
2: Define: $f_K = \log K$
3: **for** each episode $k = 1, 2, \cdots$ **do**
4:     Execute and update Algorithm 4, receive value estimators $\hat{V}_k^\pi$ for each $\pi \in \Pi$
5:     **if** $k \geq L$ and there exists a policy $\hat{\pi} \in \Pi$ such that

$$\sum_{s=1}^k Y_s \leq \sum_{s=1}^k \hat{V}_s^{\hat{\pi}} + 5\sqrt{f_K C_1 k}\,, \tag{3}$$

$$\sum_{s=1}^k Y_s \geq \sum_{s=1}^k \hat{V}_s^{\pi} + 25\sqrt{f_K C_1 k}\,, \quad \forall \pi \neq \hat{\pi}\,. \tag{4}$$

        **then**
6:         $k_0 = k, \hat{\Delta}_\pi = \frac{1}{k_0}\left(\sum_{s=1}^k \hat{V}_s^{\hat{\pi}} - \hat{V}_s^{\pi}\right), \hat{\Delta} = \left\{\left(\hat{\Delta}_\pi\right)_{\pi \in \Pi}\right\}$; break
7:     **end if**
8: **end for**
9: **for** episode $k = k_0 + 1, k_0 + 2, \cdots$ **do**
10:     Compute $p_k = \mathbf{OP}(k, \hat{\Delta})$
11:     Compute $\tilde{p}_k(\pi) = \frac{1}{2}e_{\hat{\pi}} + \frac{1}{2}p_k(\pi)$, where $e_{\hat{\pi}}$ is a one-hot vector with 1 only at policy $\hat{\pi}$
12:     Sample $\pi_k \sim \tilde{p}_k$ and execute $\pi_k$
13:     Receive rewards $Y_k = \sum_{h=1}^H y_{k,h}$ and calculate $\hat{V}_k^\pi$ for each $\pi \in \Pi$ as follows

$$\forall \pi \neq \hat{\pi} : \hat{V}_k^\pi = \sum_{h=1}^H \phi_{\pi,h}^\top \Sigma_{k,h}^{-1} \phi_{\pi_k,h} y_{k,h}\,, \text{ where } \Sigma_{k,h} = \sum_\pi \tilde{p}_k(\pi)\phi_{\pi,h}\phi_{\pi,h}^\top\,; \tag{5}$$

$$\hat{V}_k^{\hat{\pi}} = \frac{Y_k}{\tilde{p}_k(\hat{\pi})}\mathbb{1}\{\pi_k = \hat{\pi}\}\,. \tag{6}$$

14:     For each $\pi \neq \hat{\pi}$, compute $\hat{\Delta}_\pi$ as

$$\hat{\Delta}_k^\pi = \frac{1}{k}\left(\sum_{s=1}^{k_0} \hat{V}_s^\pi + (k - k_0)\mathrm{Rob}_{k,\pi} - \sum_{s=1}^k \hat{V}_s^{\hat{\pi}}\right)\,, \text{ with } \mathrm{Rob}_{k,\pi} = \mathbf{Catoni}_{\alpha_k^\pi}\left(\{\hat{V}_s^\pi\}_{s=k_0+1}^k\right) \tag{7}$$

15:     **if**

$$\exists \pi \neq \hat{\pi}\,, \hat{\Delta}_k^\pi \notin \left[0.39\hat{\Delta}_\pi, 1.81\hat{\Delta}_\pi\right] \text{ or} \tag{8}$$

$$\sum_{s=k_0+1}^k \left(\hat{V}_s^{\hat{\pi}} - Y_s\right) \geq 20\sqrt{f_K C_1 k_0} \tag{9}$$

        **then**
16:         Return $k_0$
17:     **end if**
18: **end for**

---

Is is worth noting that an adversarial setting may be disguised as stochastic scenarios and fool the algorithm to enter in the stochastic phase. Thus the agent still needs to be vigilant about the possible change of the environment. The detection conditions (Line 15) are set for this objective. To be specific, if the estimated sub-optimality gap of a policy changes obviously compared with the original estimation by Algorithm 4 in the adversarial phase or the regret compared with $\hat{\pi}$ is large, the algorithm can determine that the environment may not be stochastic and would terminiate the current epoch and enter in the next epoch with parameter $k_0$ (Line 16).

---

**Algorithm 3** Optimization problem $(\mathbf{OP})\Big(k, \hat{\Delta}\Big)$

---

1: Define $\beta_k = 2^{15} H \log\big(|\Pi| k / \delta\big)$
2: Return the minimizer $p^*$ of the following constrained optimization problem

$$\min_p \sum_{\pi \in \Pi} p_\pi \hat{\Delta}_\pi \tag{10}$$

$$\text{s.t.} \sum_{h=1}^{H} \big\| \phi_{\pi,h} \big\|^2_{\Sigma_h^{-1}(p)} \leq \frac{k \hat{\Delta}_\pi}{\beta_k} + 4dH \text{ and } \sum_{\pi \in \Pi} p_\pi = 1 \,, \tag{11}$$

where $\Sigma_h(p) = \sum_\pi p_\pi \phi_{\pi,h} \phi_{\pi,h}^\top$.

---

## 5 THEORETICAL ANALYSIS

In this section, we provide the theoretical guarantee and the analysis of Algorithm 1 in both stochastic and adversarial settings

The first is about the stochastic setting. Since the value function remains the same for different episodes, we simplify the notation and use $V^\pi$ to represent the real value of policy $\pi \in \Pi$. Before presenting the main results, we first introduce the sub-optimality gaps that will be used.

**Definition 1.** *For each policy $\pi \in \Pi$, define $\Delta_\pi = V^{\pi^*} - V^\pi$ as the sub-optimality gap of $\pi$ compared with the optimal policy $\pi^* \in \arg\max_{\pi \in \Pi} V^\pi$. Further let $\Delta_{\min} = \min_{\pi : \Delta_\pi > 0} \Delta_\pi$ be the minimum non-negative value gap.*

Theorem 2 provides a regret upper bound for Algorithm 1 in the stochastic setting.

**Theorem 2.** *(Regret bound in the stochastic setting) With probability at least $1 - \delta$, Algorithm 1 guarantees that*

$$\text{Reg}(K; \Pi) \leq \mathcal{O}\left( \frac{dH^2 \log(K) \log\big(|\Pi| K / \delta\big)}{\Delta_{\min}} \right) . \tag{12}$$

And if the environment is adversarial, the regret of Algorithm 1 can be upper bounded as Theorem 3.

**Theorem 3.** *(Regret bound in the adversarial setting) With probability at least $1 - \delta$, Algorithm 1 guarantees that*

$$\text{Reg}(K; \Pi) \leq \mathcal{O}\left( \sqrt{dH^3 K \log(K) \log\big(|\Pi| K / \delta\big)} \right) . \tag{13}$$

Due to the space limit, the full proof of these two theorems are deferred to Appendix B and C. We will provide a proof sketch for them in later sections.

**Technique challenge and novelty.** There are mainly two types of algorithms to deal with the BoBW problem: the switch-based method which actively detects the environment type, e.g., Bubeck & Slivkins (2012), and the FTRL-based method which adapts to different environments, e.g. Zimmert & Seldin (2019). The approach in Bubeck & Slivkins (2012) first assumes the setting to be stochastic and would detect whether a policy's value has changed. Such an approach in our setting brings an $O(\sqrt{|\Pi|})$ dependence in the regret for adversarial setting which is not idealistic as the policy set size can be large. And the success of FTRL for BoBW mainly relies on a self-bounding inequality that bounds the regret by the chosen probabilities of policies. But such a technique is challenging with linear structure. As discussed by Lee et al. (2021), even for the single-state linear bandit setting, connecting FTRL with **OP** is hard.

Our approach relies on a new observation that the value of each policy can be written as the inner product between the expected state-action visitation feature $\phi_\pi$ and the unknown reward feature $\theta$. From this view, we are able to reduce the problem to linear optimization and existing techniques for linear optimization can be used. To the best of our knowledge, we are the first to introduce this type of linear optimization for the regret minimization problem in MDP and such reduction may be of independent interest.

**Relationship between our $\Delta_\pi$ and the $\text{gap}_{\min}$ in He et al. (2021).** To compare our $\Delta_\pi$ with $\text{gap}_{\min}$ in He et al. (2021), we assume the optimal policy $\pi^* \in \Pi$ is just the global optimal policy. Recall that $\text{gap}_{\min}$ is defined as $\min_{h,s,a}\left\{\text{gap}_h(s,a) : \text{gap}_h(s,a) > 0\right\}$ where $\text{gap}_h(s,a) = V_h^*(s) - Q_h^*(s,a)$. In general, $\Delta_\pi$ can be decomposed as $\Delta_\pi = V_1^*(s_1) - V_1^\pi(s_1) = V_1^*(s_1) - Q_1^\pi(s_1,a') \geq V_1^*(s_1) - Q_1^*(s_1,a') = \text{gap}_1(s_1,a')$ where the first equality follows He et al. (2021, Eq. (B.2)) and $p_h^\pi(s,a)$ is the visitation probability of state-action $(s,a)$ at step $h$ by following $\pi$. This shows that $\Delta_\pi \geq p^\pi \text{gap}_{\min}$ in the worst case where $p^\pi$ is the minimum none-zero visitation probability of policy $\pi$ at some state-action pair.

And there are also cases where our defined $\Delta_\pi$ is larger than that in He et al. (2021). When both the policy and transition are deterministic (Ortner, 2010; Tranos & Proutiere, 2021; Dann et al., 2021; Tirinzoni et al., 2022), we have $\Delta_\pi = \sum_{h=1}^H \text{gap}_h(s_h,a_h) \geq \text{gap}_{\min}$. And in the stochastic transition case, if all sub-optimal policies happen to not select the optimal action in $\arg\max_a Q_1^*(s_1,a)$, $\Delta_\pi = V_1^*(s_1) - V_1^\pi(s_1) = V_1^*(s_1) - Q_1^\pi(s_1,a') \geq V_1^*(s_1) - Q_1^*(s_1,a') = \text{gap}_1(s_1,a') \geq \text{gap}_{\min}$, where $a'$ is the action selected by $\pi$ at the first step and the last inequality is due to $\text{gap}_1(s_1,a') > 0$. In the above two cases, our our sub-optimality gap is larger than previously defined gap and our dependence on the gap is better.

To the best of our knowledge, Algorithm 1 is the first that can simultaneously achieve $O(\text{poly}\log K)$ regret in the stochastic setting and $\tilde{O}(\sqrt{K})$ regret in the adversarial setting for linear MDP problem. It is also worth noting that previous works on the separate adversarial setting only provide the upper bound for the expected regret (Neu & Olkhovskaya, 2021), and we are the first to provide a high-probability guarantee.

### 5.1 REGRET ANALYSIS IN THE STOCHASTIC SETTING

In the stochastic setting, we consider the regret in two phases of each epoch separately. We first show that, the first phase (which we call as the adversarial phase, Line 3-8 in Algorithm 2) will terminate after $k_0$ episodes where $k_0 \in \left[64 f_K C_1/\Delta_{\min}^2, 900 f_K C_1/\Delta_{\min}^2\right]$ with high probability, and the optimal policy $\pi^* \in \Pi$ can be identified. Lemma 1 summarizes the formal claims.

**Lemma 1.** *In the stochastic setting, in each epoch, the following 4 claims hold.*

*1. With probability at least $1 - 4\delta$, $k_0 \leq \max\left\{\frac{900 f_K C_1}{\Delta_{\min}^2}, L\right\}$.*

*2. With probability at least $1 - \delta$, $\hat{\pi} = \pi^*$.*

*3. With probability at least $1 - 2\delta$, $k_0 \geq \frac{64 f_K C_1}{\Delta_{\min}^2}$.*

*4. With probability at least $1 - 3\delta$, $\hat{\Delta}_\pi \in \left[0.7\Delta_\pi, 1.3\Delta_\pi\right], \forall \pi \neq \pi^*$.*

The detailed proof of Lemma 1 is deferred to Appendix B. We next will give a proof sketch of Theorem 2 based on the results of Lemma 1. According to claim 1 in Lemma 1, we know that $k_0 = \mathcal{O}\left(f_K C_1/\Delta_{\min}^2\right)$, so we can bound the regret in the first phase using the guarantees of Algorithm 4 in Theorem 1 by $\sqrt{C_1 k_0} = \mathcal{O}\left(C_1 \sqrt{\log(K)}/\Delta_{\min}\right)$.

And after $k_0$ episodes, the algorithm would transfer to the second phase, which we call as the stochastic phase (Line 9-18 in Algorithm 2). If the environment is truly stochastic, the values of all policies would remain stationary and the detection condition (Line 15 in Algorithm 2) would never be satisfied. Thus the stochastic phase will not end (proved in Lemma 8 in Appendix B). As for the regret suffered in this phase, we can analyze it using the properties of **OP**. According to claim 4 in Lemma 1, the estimated sub-optimality gap is close to the real sub-optimality gap as $\hat{\Delta}_\pi \in \left[\Delta_\pi/\sqrt{3}, \sqrt{3}\Delta_\pi\right]$.

Thus performing policies based on the solution of **OP** with $\hat{\Delta}$ can reach the real instance optimality.

Specifically, we can first bridge the gap between the expected regret and the regret that occurred using Freedman inequality (Lemma 12).

$$\sum_{s=k_0+1}^{k} \Delta_{\pi_s} \leq 2\sum_{s=k_0+1}^{k}\sum_{\pi} \tilde{p}_s(\pi)\Delta_\pi + 2H\log\left(\frac{1}{\delta}\right).$$

Recall that $\tilde{p}$ is computed based on **OP** under $\hat{\Delta}$, which is close to the real sub-optimality gap $\Delta$. The regret occurred in phase 2 in episodes $k$ larger than a problem dependent constant $M =$

$\mathcal{O}\left(dH\beta_K/\Delta_{\min}^2\right)$, which is the dominating part in the expected regret, can be bounded using Lemma 7 with $r = 3$:

$$\sum_{s=M}^{k}\sum_{\pi}\tilde{p}_s(\pi)\Delta_\pi \leq \sum_{s=M}^{k}\frac{72dH\beta_s}{\Delta_{\min}s} = \mathcal{O}\left(\frac{dH\beta_k\log(k)}{\Delta_{\min}}\right).$$

Above all, we can conclude that with probability at least $1 - \delta$, the regret can be upper bounde as

$$\text{Reg}\left(K;\Pi\right) = \mathcal{O}\left(\frac{dH\beta_K\log(K)}{\Delta_{\min}}\right) = \mathcal{O}\left(\frac{dH^2\log(K)\log\left(|\Pi|K/\delta\right)}{\Delta_{\min}}\right).$$

### 5.2 REGRET ANALYSIS IN THE ADVERSARIAL SETTING

In the adversarial setting, the regret in the first phase can be guaranteed with the property of Algorithm 4 in Theorem 1. Here we will mainly analyze the second phase. We first show that in the second phase of each epoch, the returned policy $\hat{\pi}$ is actually the optimal policy in $\Pi$.

**Lemma 2.** *For any episode $k$ in the second phase, the policy $\hat{\pi}$ has the most accumulated value in $\Pi$ during episodes $1$ to $k$. That is, $\hat{\pi} \in \arg\max_{\pi\in\Pi}\sum_{\kappa=1}^{k}V_\kappa^\pi$.*

Since $\hat{\pi}$ is the optimal policy in $\Pi$, the regret can be written as the sum of the deviation between the value of the selected policy $\pi_s$ and $V_s^{\hat{\pi}}$

$$\sum_{s=k_0+1}^{k}\left(V_s^{\hat{\pi}} - V_s^{\pi_s}\right) = \sum_{s=k_0+1}^{k}\left[\left(\hat{V}_s^{\hat{\pi}} - Y_s\right) + \left(Y_s\mathbb{1}\{\pi_s = \hat{\pi}\} + V_s^{\hat{\pi}}\mathbb{1}\{\pi_s \neq \hat{\pi}\} - \hat{V}_s^{\hat{\pi}}\right)\right.$$

$$\left. + \left(\left(V_s^{\hat{\pi}}\mathbb{1}\{\pi_s = \hat{\pi}\} - Y_s\mathbb{1}\{\pi_s = \hat{\pi}\}\right) + \left(Y_s - V_s^{\pi_s}\right)\right)\right].$$

According to the detection condition (equation 9) in Algorithm 2, the first term can be upper bounded by $\mathcal{O}\left(\sqrt{f_KC_1k_0}\right)$. As for the second and last term, Freedman inequality (Lemma 12) also provides an upper bound $\mathcal{O}\left(C_1k_0\right)$ for them. Above all, the regret in a single epoch can be upper bounded by $\mathcal{O}\left(\sqrt{f_KC_1k_0}\right)$. According to the choice of the minimal duration $L$ in each epoch, the length $k_0$ of the first phase in an epoch is at most half of that in the next epoch. Thus the final regret can be bounded as

$$\text{Reg}\left(K;\Pi\right) = \mathcal{O}\left(\sqrt{C_1K\log(K)}\right) = \mathcal{O}\left(\sqrt{dH^2K\beta_K\log(K)}\right).$$

## 6 CONCLUSION

In this paper, we propose the first BoBW algorithm for linear MDP that can simultaneously achieve $O(\text{poly}\log K)$ regret in the stochastic setting and $\tilde{O}(\sqrt{K})$ regret in the adversarial setting. Our approach relies on the new observation that the value function of a policy can be written as the sum of the inner products between the expected state-action visitation feature $\phi_{\pi,h}$ and the unknown reward parameter $\theta_h$ at different steps $h \in [H]$. And the problem can thus be regarded as an online linear optimization problem.

Apart from these BoBW results, we also propose a new analysis that can reach a high-probability regret guarantee for adversarial linear MDP, which is also the first such result in the literature.

An important future direction is to remove the assumption of unique optimal policy in the stochastic setting. This assumption also appears in previous BoBW works for tabular MDP (Jin & Luo, 2020; Jin et al., 2021b) and linear bandits (Lee et al., 2021) which destroys the generality of the results but is challenging to be removed due to the hardness of the BoBW objective and the complex structure of MDP. Extending the current results to the unknown transition setting is also prospective. This is hoped to be solved by some new techniques since the current approach highly depends on the known state-action visitation feature. Deriving an FTRL-type algorithm for this objective is also an interesting future direction. It still remains open even in the simpler linear bandit setting without the transition between different states.

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

# A   ANALYSIS OF **OP**

The analysis of **OP** mainly follows the op problem for linear bandits (Lee et al., 2021) but with the consideration of the special structure of linear MDP. In this section, we provide some useful lemmas for **OP** in linear MDP.

**Lemma 3.** *First, consider the constrained optimization problem:*

$$\min_{p \in \Delta_\Pi} \sum_\pi p_\pi \hat{\Delta}_\pi - \frac{2}{\xi} \sum_{h=1}^{H} \left( -\ln(\det(\Sigma_h(p))) \right),$$

*where* $\Sigma_h(p) = \sum_\pi p_\pi \phi_{\pi,h} \phi_{\pi,h}^\top$.

*The optimal choice of* $p = p^*$ *yields that:*

$$\sum_{\pi \in \Pi} p_\pi^* \hat{\Delta}_\pi \leq \frac{2dH}{\xi},$$

$$\sum_{h=1}^{H} \|\phi_{\pi,h}\|_{\Sigma_h^{-1}(p^*)}^2 \leq \frac{\xi \hat{\Delta}_\pi}{2} + dH, \quad \forall \pi \in \Pi.$$

*Proof.* Here we relaxed the constraint that $p_\pi$ be a valid distribution on $\Pi$ to $\sum_\pi p_\pi \leq 1$ since there must be $\hat{\pi}^*$ where $\hat{\Delta}_{\hat{\pi}^*} = 0$, so we can always add up the probability on $\hat{\pi}^*$ to make it a distribution. Applying the KKT conditions and setting the derivatives with respect to each $p_\pi$ in the Lagrangian to zero, we got:

$$0 = \hat{\Delta}_\pi - \frac{2}{\xi} \sum_{h=1}^{H} \|\phi_{\pi,h}\|_{\Sigma_h^{-1}(p^*)}^2 - \lambda_\pi + \lambda,$$

where $\lambda_\pi$ and $\lambda$ are the respective Lagrange multipliers for the constraints of $p_\pi \geq 0$ and $\sum_\pi p_\pi \leq 1$, thus we have $\lambda \geq 0$ and $\sum_\pi \lambda_\pi p_\pi = 0$. Multiplying the above equation with $p_\pi^*$ and summing over $\pi \in \Pi$, we got:

$$0 = \sum_{\pi \in \Pi} \left( p_\pi^* \hat{\Delta}_\pi - \frac{2}{\xi} \sum_{h=1}^{H} p_\pi^* \|\phi_{\pi,h}\|_{\Sigma_h^{-1}(p^*)}^2 - \lambda_\pi p_\pi^* + \lambda p_\pi^* \right)$$

$$= \sum_{\pi \in \Pi} p_\pi^* \hat{\Delta}_\pi - \frac{2}{\xi} \sum_{h=1}^{H} \sum_{\pi \in \Pi} p_\pi^* \|\phi_{\pi,h}\|_{\Sigma_h^{-1}(p^*)}^2 + \lambda.$$

Notice that:

$$\sum_{\pi \in \Pi} \|\phi_{\pi,h} p_\pi^*\|_{\Sigma_h^{-1}(p^*)}^2 = \sum_{\pi \in \Pi} p_\pi^* \phi_{\pi,h} \Sigma_h^{-1}(p^*) \phi_{\pi,h}^\top$$

$$= \sum_{\pi \in \Pi} \text{Tr} \left( p_\pi^* \phi_{\pi,h} \phi_{\pi,h}^\top \Sigma_h^{-1}(p^*) \right)$$

$$= \text{Tr} \left( \sum_{\pi \in \Pi} p_\pi^* \phi_{\pi,h} \phi_{\pi,h}^\top \Sigma_h^{-1}(p^*) \right)$$

$$= d.$$

Plugging in this result, we got:

$$0 = \sum_{\pi \in \Pi} p_\pi^* \hat{\Delta}_\pi - \frac{2}{\xi} dH + \lambda$$

$$\geq \sum_{\pi \in \Pi} p_\pi^* \hat{\Delta}_\pi - \frac{2}{\xi} dH.$$

So that

$$\sum_{\pi \in \Pi} p_\pi^* \hat{\Delta}_\pi \leq \frac{2}{\xi} dH \,,$$

and $\lambda \leq \frac{2}{\xi} dH$, since $\sum_{\pi \in \Pi} p_\pi^* \hat{\Delta}_\pi \geq 0$.
Above all,

$$\sum_{h=1}^{H} \|\phi_{\pi,h}\|_{\Sigma_h^{-1}(p^*)}^2 = \frac{\xi}{2}(\hat{\Delta}_\pi - \lambda_\pi + \lambda) \leq \frac{\xi}{2}(\hat{\Delta}_\pi + \lambda) \leq \frac{\xi \hat{\Delta}_\pi}{2} + dH \,.$$

$\square$

**Lemma 4.** *Suppose that for any $h \in \{1, 2, \cdots, H\}$, $\{\phi_{\pi,h} \mid \pi \in \Pi\}$ spans $R^d$. Denote $p_\Pi$ as a uniform distribution on $\Pi$ and let $\kappa \in (0, \frac{1}{2})$. For any $G \subset \Pi$, there exists distribution on $q \in P_G$ such that $\sum_{h=1}^{H} \|\phi_{\pi,h}\|_{\Sigma_h^{-1}(q^{G,\kappa})}^2 \leq 2dH$, where $q^{G,\kappa} = \kappa p_\Pi + (1 - \kappa)q$ and $\Sigma_h(q^{G,\kappa}) = \sum_{\pi \in \Pi} q_\pi^{G,\kappa} \phi_{\pi,h} \phi_{\pi,h}^\top$.*

*Proof.* Denote $P^{G,\kappa} = \{\kappa p_\Pi + (1 - \kappa)q \mid q \in P_G\}$.

$$\min_{q \in P^{G,\kappa}} \max_{\pi \in G} \sum_{h=1}^{H} \|\phi_{\pi,h}\|_{\Sigma_h^{-1}(q)}^2$$

$$= \min_{q \in P^{G,\kappa}} \max_{p \in P_G} \sum_{h=1}^{H} \mathrm{Tr} \left( \sum_{\pi \in G} p_\pi \phi_{\pi,h} \phi_{\pi,h}^\top \right) \left( \sum_{\pi \in \Pi} q_\pi \phi_{\pi,h} \phi_{\pi,h}^\top \right)^{-1}$$

$$= \max_{p \in P_G} \min_{q \in P^{G,\kappa}} \sum_{h=1}^{H} \mathrm{Tr} \left( \sum_{\pi \in G} p_\pi \phi_{\pi,h} \phi_{\pi,h}^\top \right) \left( \sum_{\pi \in \Pi} q_\pi \phi_{\pi,h} \phi_{\pi,h}^\top \right)^{-1} \quad (14)$$

$$\leq \max_{p \in P_G} \sum_{h=1}^{H} \mathrm{Tr} \left[ (\sum_{\pi \in G} p_\pi \phi_{\pi,h} \phi_{\pi,h}^\top) \left( \sum_{\pi \in \Pi} \left( \frac{\kappa}{|\Pi|} + (1 - \kappa)p_\pi \right) \phi_{\pi,h} \phi_{\pi,h}^\top \right)^{-1} \right]$$

$$\leq 2 \max_{p \in P_G} \sum_{h=1}^{H} \mathrm{Tr} \left[ \left( \sum_{\pi \in \Pi} \left( \frac{\kappa}{|\Pi|} + (1 - \kappa)p_\pi \right) \phi_{\pi,h} \phi_{\pi,h}^\top \right) \left( \sum_{\pi \in \Pi} \left( \frac{\kappa}{|\Pi|} + (1 - \kappa)p_\pi \right) \phi_{\pi,h} \phi_{\pi,h}^\top \right)^{-1} \right]$$

$$= 2dH \,.$$

Where the second inequality is due to Sion's minimax theorem as equation 14 is linear in $p$ and convex in $q$. The last inequality is due to $1 - \kappa \geq \frac{1}{2}$.

$\square$

**Lemma 5.** *Let $p_\pi$ be the solution of $\textbf{OP}(k, \hat{\Delta})$, then we have :*

$$\sum_{\pi \in \Pi} p_\pi \hat{\Delta}_\pi \leq \mathcal{O}\left( \frac{dH\beta_k}{\sqrt{k}} \right) \,. \quad (15)$$

*Proof.* We transform $p^*$ in Lemma 3 to a solution satisfying $\textbf{OP}$.
Choosing $\xi$ as $\frac{\sqrt{k}}{\beta_k}$ in Lemma 4, and let $G = \{\pi \in \Pi : \hat{\Delta}_\pi \leq \frac{1}{\sqrt{k}}\}$, construct the distribution $q = \frac{1}{2}p^* + \frac{1}{2}q^{G,\kappa}$, where $q^{G,\kappa}$ is the distribution stated above with $\kappa = \frac{1}{\sqrt{k}}$ in Lemma 4, we have for $\pi \in G$,

$$\sum_{h=1}^{H} \|\phi_{\pi,h}\|_{\Sigma_h^{-1}(q)}^2 \leq 4dH;$$

and for $\pi \notin G$,

$$\sum_{h=1}^{H} \|\phi_{\pi,h}\|_{\Sigma_h^{-1}(q)}^2 \le 2 \left( \frac{\xi \hat{\Delta}_\pi}{2} + dH \right) \le \frac{\sqrt{k}\hat{\Delta}_\pi}{\beta_k} + 2dH \le \frac{k\hat{\Delta}_\pi^2}{\beta_k} + 4dH .$$

So distribution $q$ satisfy the constraints of **OP**.
For the optimal solution $p$ of **OP**, we have:

$$\sum_{\pi \in \Pi} p_\pi \hat{\Delta}_\pi$$

$$\le q_\pi \hat{\Delta}_\pi = \left( \frac{1}{2} p^* + \frac{1}{2} q^{G,\kappa} \right) \hat{\Delta}_\pi$$

$$\le \frac{dH\beta_k}{\sqrt{k}} + \frac{H}{2\sqrt{k}} + \frac{1}{2\sqrt{k}} = \mathcal{O}\left( \frac{dH\beta_k}{\sqrt{k}} \right) .$$

$\square$

**Lemma 6.** *Given* $\{\hat{\Delta}_\pi\}_{\pi \in \Pi}$, *suppose there exists unique* $\hat{\pi}$ *such that* $\hat{\Delta}_{\hat{\pi}} = 0$, *and* $\hat{\Delta}_{\min} = \min_{\hat{\Delta}_\pi > 0} \hat{\Delta}_\pi$. *Let* $p_\pi$ *be the solution of* **OP**$(k, \Delta)$, *when* $k \ge \frac{16dH\beta_k}{\hat{\Delta}_{\min}^2}$, *we have* $\sum_{\pi \in \Pi} p_\pi \hat{\Delta}_\pi \le \frac{24dH\beta_k}{\hat{\Delta}_{\min}k}$.

*Proof.* let $G_i = \{\pi \in \Pi : 2^{i-1}\hat{\Delta}_{\min}^2 \le \hat{\Delta}_\pi^2 \le 2^i \hat{\Delta}_{\min}^2\}$ and $n$ be the largest index that $G_i$ is not empty. Define $z_i = \frac{dH\beta_k}{2^{i-2}\hat{\Delta}_{\min}^2 k}$ and $\kappa = \frac{1}{n2^n}$. Define the distribution $\tilde{p}$ as follows:
For $\pi \ne \hat{\pi}$,

$$\tilde{p}_\pi = \sum_{i \ge 1} z_i q_\pi^{G_i, \kappa} ;$$

for $\hat{\pi}$,

$$\tilde{p}_{\hat{\pi}} = 1 - \sum_{\pi \ne \hat{\pi}} \tilde{p}_\pi .$$

We show it's a valid distribution over $\Pi$:

$$\tilde{p}_{\hat{\pi}} = 1 - \sum_{\pi \ne \hat{\pi}} \tilde{p}_\pi$$

$$= 1 - \sum_{\pi \ne \hat{\pi}} \sum_i z_i q_\pi^{G_i, \kappa}$$

$$\ge 1 - \sum_{i \ge 1} z_i - \sum_{i \ge 1} \sum_{\pi \in G_i} \sum_{j \ne i} z_j q_\pi^{G_j, \kappa}$$

$$= 1 - \sum_{i \ge 1} z_i - \sum_{i \ge 1} \sum_{\pi \in G_i} \sum_{j \ne i} \frac{z_j}{n2^n |\Pi|}$$

$$\ge 1 - 2 \sum_{i \ge 1} z_i$$

$$\ge \frac{1}{2} .$$

Where the last inequality is due to $k \ge \frac{16dH\beta_k}{\hat{\Delta}_{\min}^2}$.
For $\pi \ne \hat{\pi}$ and $\pi \in G_i$,

$$\sum_{h=1}^{H} \|\phi_{\pi,h}\|_{\Sigma_h^{-1}(\tilde{p})}^2 \le \sum_{h=1}^{H} \|\phi_{\pi,h}\|_{\left( \sum_{\pi \in \Pi} z_i q_\pi^{G_i,\kappa} \phi_{\pi,h} \phi_{\pi,h}^\top \right)^{-1}}^2 \le \frac{2dH}{z_i} \le \frac{k\hat{\Delta}_\pi^2}{\beta_k} + 4dH .$$

For $\hat{\pi}$,

$$\sum_{h=1}^{H}\left\|\phi_{\hat{\pi},h}\right\|_{\Sigma_h^{-1}(\tilde{p})}^2$$

$$\geq \sum_{h=1}^{H}\left\|\Sigma_h^{-1}(\tilde{p})\phi_{\hat{\pi},h}\right\|_{\Sigma_h^{-1}(\tilde{p})}^2$$

$$\geq \sum_{h=1}^{H}\left\|\Sigma_h^{-1}(\tilde{p})\phi_{\hat{\pi},h}\right\|_{\left(\frac{1}{2}\phi_{\hat{\pi},h}\phi_{\hat{\pi},h}^\top\right)}^2$$

$$\geq \frac{1}{2}\sum_{h=1}^{H}\left\|\phi_{\hat{\pi},h}\right\|_{\Sigma_h^{-1}(\tilde{p})}^4$$

$$\geq \frac{1}{2H}\left(\sum_{h=1}^{H}\left\|\phi_{\hat{\pi},h}\right\|_{\Sigma_h^{-1}(\tilde{p})}^2\right)^2,$$

where the last inequality is due to Cauchy inequality.
So that:

$$\sum_{h=1}^{H}\left\|\phi_{\hat{\pi},h}\right\|_{\Sigma_h^{-1}(\tilde{p})}^2 \leq 2H.$$

Thus, $\tilde{p}$ satisfy the constraints of **OP**.
Now we bound the result of **OP**. By the optimality of $p_\pi$:

$$\sum_{\pi\in\Pi}p_\pi\hat{\Delta}_\pi \leq \sum_{\pi\in\Pi}\tilde{p}_\pi\hat{\Delta}_\pi$$

$$= \sum_{i\geq 1}\sum_{\pi\in G_i}\left(z_i q^{G_i,\kappa}+\sum_{j\neq i}z_j\frac{1}{n2^n|\Pi|}\right)2^{\frac{i}{2}}\hat{\Delta}_{\min}$$

$$\leq \sum_{i\geq 1}\sum_{\pi\in G_i}\sum_{j\neq i}\frac{dH\beta_k}{n2^{n+j-\frac{i}{2}-2}|\Pi|\hat{\Delta}_{\min}k}+\sum_{i\geq 1}\frac{dH\beta_k}{2^{\frac{i}{2}-2}\hat{\Delta}_{\min}k}$$

$$\leq 2\sum_{i\geq 1}\frac{dH\beta_k}{2^{\frac{i}{2}-2}\hat{\Delta}_{\min}k} \leq \frac{24dH\beta_k}{\hat{\Delta}_{\min}k}.$$

□

**Lemma 7.** *Suppose that $\hat{\Delta}_\pi \in \left(\frac{1}{\sqrt{r}}\Delta_\pi, \sqrt{r}\Delta_\pi\right)$, then the solution $p_\pi$ of **OP**$(k,\hat{\Delta})$ for $k \geq \frac{16rdH\beta_k}{\Delta_{\min}^2}$ yields that :*

$$\sum_{\pi\in\Pi}p_\pi\Delta_\pi \leq \frac{24rdH\beta_k}{\Delta_{\min}k}.$$

*Proof.* By the condition on $\Delta_\pi$, we have $t \geq \frac{16rdH\beta_k}{\Delta_{\min}^2} \geq \frac{16dH\beta_k}{\hat{\Delta}_{\min}^2}$, and that $\Delta_{\pi^*}=\hat{\Delta}_{\pi^*}=0$. Thus,

$$\sum_{\pi\in\Pi}p_\pi\Delta_\pi \leq \sqrt{r}\sum_{\pi\in\Pi}p_\pi\hat{\Delta}_\pi \leq \sqrt{r}\frac{24dH\beta_k}{\hat{\Delta}_{\min}k} \leq \frac{24rdH\beta_k}{\Delta_{\min}k}.$$

Where the inequality is due to Lemma 6. □

## B  ANALYSIS IN THE STOCHASTIC SETTING

*Proof of Lemma 1.* First, we show the following properties, for any $k$ in phase 1:

$$C_2\left|\sum_{s=1}^{k}\hat{V}_s^\pi - V_s^\pi\right| \leq \sqrt{C_1 k}+\sum_{s=1}^{k}V_s^{\pi_s}-V_s^\pi \leq \sqrt{C_1 k}+k\Delta_\pi.$$

Here we denote $\text{DEV}_{k,\pi} = \left| \sum_{s=1}^{k} \hat{V}_s^{\pi} - V_s^{\pi} \right|$.

**Claim 1's proof:** Let $k = \max\{ \frac{900 f_K C_1}{\Delta_{\min}^2}, L \}$ and assume that phase 1 has not finished at episode $k$. Set $\hat{\pi} = \pi^*$, and we show that the termination conditions hold with high probability at episode $k$. According to the Azuma-Hoeffding inequality, since $Y_s - V_s^{\pi_s}$ is a martingale sequence and that $|Y_s - V_s^{\pi_s}| \leq H$,

$$\sum_{s=1}^{k} Y_s \leq \sum_{s=1}^{k} V_s^{\pi_s} + \sqrt{C_1 k}$$

$$\leq \sum_{s=1}^{K} V_s^{\pi^*} + \sqrt{C_1 k}$$

$$\leq \sum_{s=1}^{K} \hat{V}_s^{\pi^*} + 2\sqrt{C_1 k}$$

$$\leq \sum_{s=1}^{K} \hat{V}_s^{\pi^*} + 3\sqrt{f_K C_1 k},$$

so equation 3 is satisfied.

For all $\pi \neq \pi^*$, we have:

$$\sum_{s=1}^{k} \hat{V}_s^{\pi} - Y_s = \sum_{s=1}^{k} \left( \hat{V}_s^{\pi} - V_s^{\pi} \right) + \left( V_s^{\pi} - V_s^{\pi^*} \right) + \left( V_s^{\pi^*} - V_s^{\pi_s} \right) + (V_s^{\pi_s} - Y_s)$$

$$\leq \text{DEV}_{k,\pi} - k\Delta_{\pi} + \sqrt{C_1 k} - C_2 \text{DEV}_{k,\pi^*} + \sqrt{C_1 k}$$

$$\leq 2\sqrt{f_K C_1 k} + \frac{1}{C_2} \left( \sqrt{C_1 k} + k\Delta_{\pi} \right) - k\Delta_{\pi}$$

$$\leq -0.95 k\Delta_{\pi} + 2.1\sqrt{f_K C_1 k}.$$

Since $k \geq \frac{900 f_K C_1}{\Delta_{\pi}^2}$, thus $k\Delta_{\pi} \geq 30\sqrt{f_K C_1 k}$ for all $\pi \neq \pi^*$. So $-0.95 k\Delta_{\pi} + 2.1\sqrt{f_K C_1 k} \leq -25\sqrt{f_K C_1 k}$.
Thus:

$$\sum_{s=1}^{k} Y_s \geq \sum_{s=1}^{k} \hat{V}_s^{\pi} + 25\sqrt{f_K C_1 k}.$$

So equation 4 is satisfied.

**Claim 2's proof:** Using equation 3 and equation 4, we got:

$$\sum_{s=1}^{k_0} \hat{V}_s^{\hat{\pi}} - \hat{V}_s^{\pi} \geq 20\sqrt{f_K C_1 k_0}, \quad \forall \pi \neq \hat{\pi}. \tag{16}$$

However, with probability at least $1 - \delta$, for any $\pi \neq \pi^*$

$$\sum_{s=1}^{k_0} \hat{V}_s^{\pi} - \hat{V}_s^{\pi^*} \leq \sum_{s=1}^{k_0} \left( \hat{V}_s^{\pi} - V_s^{\pi} \right) + \left( V_s^{\pi} - V_s^{\pi^*} \right) + \left( V_s^{\pi^*} - \hat{V}_s^{\pi^*} \right)$$

$$\leq \text{DEV}_{k_0,\pi} + \text{DEV}_{k_0,\pi^*} - k_0 \Delta_{\pi}$$

$$\leq \frac{1}{C_2} \left( \sqrt{C_1 k_0} + k_0 \Delta_{\pi} \right) + \frac{1}{C_2} \sqrt{C_1 k_0} - k_0 \Delta_{\pi}$$

$$\leq 5\sqrt{f_K C_1 k_0}.$$

So we must have $\hat{\pi} = \pi^*$.

**Claim 3's proof:** Suppose that $k_0 \leq \frac{64 f_K C_1}{\Delta_{\min}^2}$. Let $\pi$ be the policy with minimal estimated gap, that is: $\Delta_\pi = \Delta_{\min}$.

$$
\begin{aligned}
\sum_{s=1}^{k_0} \hat{V}_s^{\pi^*} - \hat{V}_s^\pi &\leq k_0 \Delta_{\min} + \mathrm{DEV}_{k_0,\pi} + \mathrm{DEV}_{k_0,\pi^*} \\
&\leq k_0 \Delta_{\min} + \frac{1}{C_2}\left(2\sqrt{C_1 k_0} + k_0 \Delta_{\min}\right) \\
&\leq 2 k_0 \Delta_{\min} + 2\sqrt{f_K C_1 k_0} \\
&\leq 16 \sqrt{f_K C_1 k_0} + 2\sqrt{f_K C_1 k_0} \\
&= 18\sqrt{f_K C_1 k_0}\, .
\end{aligned}
$$

Which contradicts with equation 16.

**Claim 4's proof:**

$$
\begin{aligned}
\left|k_0 \hat{\Delta}_\pi - k_0 \Delta_\pi\right| &\leq \mathrm{DEV}_{k_0,\pi} + \mathrm{DEV}_{k_0,\pi^*} \\
&\leq \frac{1}{C_2}\left(2\sqrt{C_1 k_0} + k_0 \Delta_\pi\right) \\
&\leq \sqrt{f_K C_1 k_0} + \frac{1}{C_2} k_0 \Delta_\pi \\
&\leq 0.3 k_0 \Delta_\pi\, .
\end{aligned}
$$

The last inequality is due to $k_0 \geq \frac{64 f_K C_1}{\Delta_{\min}^2}$.
So we have:

$$
\hat{\Delta}_\pi \in [0.7\Delta_\pi, 1.3\Delta_\pi]\, , \quad \forall \pi \neq \pi^*\, .
$$

$\square$

**Lemma 8.** *With probability at least $1 - \delta$, phase 2 never ends.*

*Proof.* For equation 9, we decompose it as

$$
\begin{aligned}
\sum_{s=k_0+1}^{k} \hat{V}_s^{\hat{\pi}} - Y_s &= \sum_{s=k_0+1}^{k}\left(V_s^{\hat{\pi}} - V_s^{\pi_s}\right) + \left(V_s^{\pi_s} - Y_s + \frac{Y_s - V_s^{\hat{\pi}}}{\tilde{p}_s(\hat{\pi})}\mathbb{1}\{\pi_s = \hat{\pi}\}\right) \\
&\quad + \left(\frac{V_s^{\hat{\pi}}}{\tilde{p}_s(\hat{\pi})}\mathbb{1}\{\pi_s = \hat{\pi}\} - V_s^{\hat{\pi}}\right)\, .
\end{aligned}
$$

First we deal with the second term, which is a martingale difference sequence. It's variance is bounded as:

$$
\begin{aligned}
&\mathbb{E}\left[\left(V_s^{\pi_s} - Y_s + \frac{Y_s - V_s^{\hat{\pi}}}{\tilde{p}_s(\pi)}\mathbb{1}\{\pi_s = \hat{\pi}\}\right)^2\right] \\
&= \tilde{p}_s(\hat{\pi})\mathbb{E}\left[(V_s^{\pi_s} - Y_s)^2\left(1 - \frac{1}{\tilde{p}_s(\hat{\pi})}\right)^2\right] + (1 - \tilde{p}_s(\hat{\pi}))\,\mathbb{E}\left[(V_s^{\pi_s} - Y_s)^2\right] \\
&\leq 2H^2\left(1 - \tilde{p}_s(\hat{\pi})\right)\, .
\end{aligned}
$$

The third term is also a martingale difference sequence, whose variance can be bounded as:

$$
\mathbb{E}\left[\left(\frac{V_s^{\hat{\pi}}}{\tilde{p}_s(\hat{\pi})}\mathbb{1}\{\pi_s = \hat{\pi}\} - V_s^{\hat{\pi}}\right)^2\right] \leq 2H^2\left(1 - \tilde{p}_s(\hat{\pi})\right)\, .
$$

Thus, using the Freedman inequality for the second and third term, we got:

$$\sum_{s=k_0+1}^{k} \left( V_s^{\pi_s} - Y_s + \frac{Y_s - V_s^{\hat{\pi}}}{\tilde{p}_s(\hat{\pi})} \mathbb{1}\{\pi_s = \hat{\pi}\} \right) + \left( \frac{V_s^{\hat{\pi}}}{\tilde{p}_s(\hat{\pi})} \mathbb{1}\{\pi_s = \hat{\pi}\} - V_s^{\hat{\pi}} \right)$$

$$\leq 8H \sqrt{\sum_{s=k_0+1}^{k} \left( 1 - \tilde{p}_s(\hat{\pi}) \right) \log \left( \frac{K}{\delta} \right)} + 6H \log \left( \frac{K}{\delta} \right).$$

For the first term, we bound it's variance against it's expectation $\sum_{\pi \neq \hat{\pi}} \tilde{p}_s(\pi) \left( V_s^{\hat{\pi}} - V_s^{\pi} \right)$ as follows:

$$\mathbb{E}\left[ \left( V_s^{\hat{\pi}} - V_s^{\pi_s} - \sum_{\pi \neq \hat{\pi}} \tilde{p}_s(\pi) \left( V_s^{\hat{\pi}} - V_s^{\pi} \right) \right)^2 \right]$$

$$= \tilde{p}_s(\hat{\pi}) \mathbb{E}\left[ \left( \sum_{\pi \neq \hat{\pi}} \tilde{p}_s(\pi) \left( V_s^{\hat{\pi}} - V_s^{\pi} \right) \right)^2 \right] + \left( 1 - \tilde{p}_s(\hat{\pi}) \right) \mathbb{E}\left[ \left( V_s^{\hat{\pi}} - V_s^{\pi_s \neq \hat{\pi}} - \sum_{\pi \neq \hat{\pi}} \tilde{p}_s(\pi) \left( V_s^{\hat{\pi}} - V_s^{\pi} \right) \right)^2 \right]$$

$$\leq \tilde{p}_s(\hat{\pi}) H^2 \left( 1 - \tilde{p}_s(\hat{\pi}) \right)^2 + \left( 1 - \tilde{p}_s(\hat{\pi}) \right) H^2 \left( 2 - \tilde{p}_s(\hat{\pi}) \right)^2$$

$$\leq 4H^2 \left( 1 - \tilde{p}_s(\hat{\pi}) \right).$$

Using Freedman inequality for the martingale difference sequence above:

$$\sum_{s=k_0+1}^{k} V_s^{\hat{\pi}} - V_s^{\pi_s}$$

$$\leq \sum_{s=k_0+1}^{k} \sum_{\pi \neq \hat{\pi}} \tilde{p}_s(\pi) \left( V_s^{\hat{\pi}} - V_s^{\pi} \right) + 4H \sqrt{\sum_{s=k_0+1}^{k} \left( 1 - \tilde{p}_s(\hat{\pi}) \right) \log \left( \frac{K}{\delta} \right)} + 2H \log \left( \frac{K}{\delta} \right)$$

$$\leq \sum_{s=k_0+1}^{k} \sum_{\pi \neq \hat{\pi}} \tilde{p}_s(\pi) \left( \Delta_{\pi} - \Delta_{\hat{\pi}} \right) + 4H \sqrt{\sum_{s=k_0+1}^{k} \left( 1 - \tilde{p}_s(\hat{\pi}) \right) \log \left( \frac{K}{\delta} \right)} + 2H \log \left( \frac{K}{\delta} \right) \tag{17}$$

$$\leq \frac{1}{2} \sum_{s=k_0+1}^{k} \sum_{\pi \neq \hat{\pi}} p_s(\pi) \Delta_{\pi} + 4H \sqrt{\sum_{s=k_0+1}^{k} \left( 1 - \tilde{p}_s(\hat{\pi}) \right) \log \left( \frac{K}{\delta} \right)} + \log \left( \frac{K}{\delta} \right),$$

where the last inequality is conditioned on that $\pi^* = \hat{\pi}$.

Using Lemma 7 and equation 31, we have:

$$\frac{1}{2} \sum_{\pi \neq \hat{\pi}} p_s(\pi) \Delta_{\pi} \leq \frac{36 d H \beta_s}{\Delta_{\min} s}, \tag{18}$$

$$1 - \tilde{p}_s(\hat{\pi}) \leq \frac{12 d H \beta_s}{\hat{\Delta}_{\min}^2 s}. \tag{19}$$

Finally, we have:

$$\sum_{s=k_0+1}^{k} \hat{V}_s^{\hat{\pi}} - Y_s$$

$$\leq \frac{36 d H \beta_k \log(k)}{\Delta_{\min}} + 12H \sqrt{\sum_{s=k_0+1}^{k} \frac{12 d H \beta_s}{\hat{\Delta}_{\min}^2 s} \log \left( \frac{K}{\delta} \right)} + 8H \log \left( \frac{K}{\delta} \right)$$

$$\leq \frac{52 d H \beta_k \log(k)}{\hat{\Delta}_{\min}} + 72 \frac{d H \log(k)}{\hat{\Delta}_{\min}} \frac{\beta_K}{\sqrt{2^{15}}} \tag{20}$$

$$\leq 20 d H \beta_K \log(k) \sqrt{\frac{k_0}{f_K C_1}} \tag{21}$$

$$\leq 20 \sqrt{f_K C_1 k_0},$$

where inequality 21 is due to claim 3 that $k_0 \geq \frac{64 f_K C_1}{\Delta_{\min}^2} \geq \frac{37 f_K C_1}{\hat{\Delta}_{\min}^2}$, inequality 20 is due to $\hat{\Delta} \in [0.7\Delta, 1.3\Delta]$ which is claim 4; and $f_K C_1 \geq dH^2 \beta_K \log(k)$.

So equation 9 is never satisfied.

According to equation 28 and equation 30, we have:

$$\left| \sum_{s=1}^{k} V_s^{\pi} - \sum_{s=1}^{k_0} \hat{V}_s^{\pi} - (k - k_0)\mathrm{Rob}_{k,\pi} \right| \leq \frac{1.4 k \hat{\Delta}_{\pi}}{10} ,$$

$$\left| \sum_{s=1}^{k} \hat{V}_s^{\hat{\pi}} - V_s^{\hat{\pi}} \right| \leq 2\sqrt{f_K C_1 k} \leq 0.1 k \hat{\Delta}_{\pi} .$$

Where the last inequality is due to equation 27.

So:

$$\left| k \hat{\Delta}_{k,\pi} - k \Delta_{\pi} \right| \leq \left| \sum_{s=1}^{k} V_s^{\pi} - \sum_{s=1}^{k_0} \hat{V}_s^{\pi} - (k - k_0)\mathrm{Rob}_{k,\pi} \right| + \left| \sum_{s=1}^{k} \hat{V}_s^{\hat{\pi}} - V_s^{\hat{\pi}} \right| \leq 0.24 k \hat{\Delta}_{\pi} .$$

Thus:

$$\hat{\Delta}_{k,\pi} \leq \Delta_{\pi} + 0.24 \hat{\Delta}_{\pi} \leq \frac{1}{0.7} \hat{\Delta}_{\pi} + 0.24 \hat{\Delta}_{\pi} \leq 1.81 \hat{\Delta}_{\pi} ,$$

$$\hat{\Delta}_{k,\pi} \geq \Delta_{\pi} - 0.24 \hat{\Delta}_{\pi} \geq \frac{1}{1.3} \hat{\Delta}_{\pi} - 0.24 \hat{\Delta}_{\pi} \geq 0.39 \hat{\Delta}_{\pi} .$$

So equation 8 is never satisfied. $\qquad \square$

*Proof of Theorem 2.* Now, we bound the deviation between the actual regret and the real regret in phase 2. Using Freedman inequality on the martingale difference sequence $\Delta_{\pi_s} - \sum_{\pi} \tilde{p}_s(\pi)\Delta_{\pi}$:

$$
\begin{aligned}
\sum_{s=k_0+1}^{k} \Delta_{\pi_s} &\leq \sum_{s=k_0+1}^{k} \sum_{\pi} \tilde{p}_s(\pi)\Delta_{\pi} + 2\sqrt{\log\left(\frac{1}{\delta}\right) \sum_{s=k_0+1}^{k} \mathbb{E}\left[\Delta_{\pi_s}^2\right]} + H \log\left(\frac{1}{\delta}\right) \\
&\leq \sum_{s=k_0+1}^{k} \sum_{\pi} \tilde{p}_s(\pi)\Delta_{\pi} + 2\sqrt{\log\left(\frac{1}{\delta}\right) H \sum_{s=k_0+1}^{k} \mathbb{E}\left[\Delta_{\pi_s}\right]} + H \log\left(\frac{1}{\delta}\right) \\
&\leq \sum_{s=k_0+1}^{k} \sum_{\pi} \tilde{p}_s(\pi)\Delta_{\pi} + 2\sqrt{\log\left(\frac{1}{\delta}\right) H \sum_{s=k_0+1}^{k} \sum_{\pi} \tilde{p}_s(\pi)\Delta_{\pi}} + H \log\left(\frac{1}{\delta}\right) \\
&\leq 2\sum_{s=k_0+1}^{k} \sum_{\pi} \tilde{p}_s(\pi)\Delta_{\pi} + 2H \log\left(\frac{1}{\delta}\right) .
\end{aligned}
\tag{22}
$$

Denote $M$ be the episode that first satisfy $M \geq \frac{48 dH\beta_M}{\Delta_{\min}^2}$.

For $k \geq M$, we have:

$$
\begin{aligned}
\sum_{s=M}^{k} \sum_{\pi} \tilde{p}_s(\pi)\Delta_{\pi} &= \sum_{s=M}^{k} \sum_{\pi} \frac{1}{2} p_s(\pi)\Delta_{\pi} \\
&\leq \sum_{s=M}^{k} \frac{36 dH\beta_s}{\Delta_{\min} s} \\
&\leq \mathcal{O}\left( \frac{dH\beta_k \log(k)}{\Delta_{\min}} \right) ,
\end{aligned}
\tag{23}
$$

where the inequality is due to Lemma 7 with $r = 3$.

For $k < M$, according to Lemma 5 we have:

$$\sum_{s=k_0+1}^{M} \sum_{\pi} \tilde{p}_s(\pi)\Delta_{\pi} = \sum_{s=k_0+1}^{M} \sum_{\pi} \frac{1}{2}p_s(\pi)\Delta_{\pi}$$

$$\leq \sum_{s=k_0+1}^{M} \mathcal{O}\left(\frac{dH\beta_s}{\sqrt{s}}\right)$$

$$\leq \mathcal{O}\left(dH\beta_M\sqrt{M}\right).$$

During phase 1, we have:

$$\sum_{s=1}^{k_0} V_s^{\pi^*} - V_s^{\pi_s} \leq \sqrt{C_1\frac{900 f_K C_1}{\Delta_{\min}^2}} \leq \mathcal{O}\left(\frac{C_1\sqrt{\log(K)}}{\Delta_{\min}}\right).$$

Since we condition on that phase 2 never ends, we have:

$$\sum_{s=1}^{K} \Delta_{\pi_s} \leq \mathcal{O}\left(\frac{C_1\sqrt{\log(K)}}{\Delta_{\min}}\right) + \mathcal{O}\left(\frac{dH\beta_K\log(K)}{\Delta_{\min}}\right) + \mathcal{O}\left(dH\beta_M\sqrt{M}\right)$$

$$\leq \mathcal{O}\left(\frac{dH^2\log(K)\log\left(\frac{|\Pi|K}{\delta}\right)}{\Delta_{\min}}\right).$$

$\square$

## C  ANALYSIS IN THE ADVERSARIAL SETTING

*Proof of Lemma 2.* First, we bound the deviation of estimation in Phase 1. For any episode k in phase 1, we have:

$$\sum_{s=1}^{k} V_s^{\pi} - V_s^{\pi_s} \leq \sqrt{C_1 K} - C_2 \left|\sum_{s=1}^{k} \hat{V}_s^{\pi} - V_s^{\pi}\right|$$

$$\leq \sqrt{C_1 K} - (C_2 - 1)\left|\sum_{s=1}^{k} \hat{V}_s^{\pi} - V_s^{\pi}\right| + \sum_{s=1}^{k}\left(V_s^{\pi} - \hat{V}_s^{\pi}\right).$$

Thus:

$$\left|\sum_{s=1}^{k} \hat{V}_s^{\pi} - V_s^{\pi}\right| \leq \frac{1}{C_2 - 1}\left(\sqrt{C_1 K} + \sum_{s=1}^{k} V_s^{\pi_s} - \hat{V}_s^{\pi}\right).$$

At time $k_0$:

$$\left|\sum_{s=1}^{k_0} \hat{V}_s^{\pi} - V_s^{\pi}\right| \leq \frac{1}{C_2 - 1}\left(\sqrt{C_1 K} + \sum_{s=1}^{k_0} V_s^{\pi_s} - \hat{V}_s^{\pi}\right)$$

$$\leq \frac{1}{C_2 - 1}\left(2\sqrt{C_1 K} + \sum_{s=1}^{k_0} Y_s - \hat{V}_s^{\pi}\right) \tag{24}$$

$$\leq \frac{1}{C_2 - 1}\left(2\sqrt{C_1 K} + 5\sqrt{f_K C_1 k_0} + \sum_{s=1}^{k_0} \hat{V}_s^{\hat{\pi}} - \hat{V}_s^{\pi}\right)$$

$$\leq \frac{1}{C_2 - 1}\left(7\sqrt{f_K C_1 k_0} + k_0\hat{\Delta}_{\pi}\right).$$

Next, we bound the deviation of $(k - k_0)\mathrm{Rob}_{k,\pi}$ for $\pi \neq \hat{\pi}$.

The variance of $\hat{V}_\kappa^\pi$ is bounded as follows:

$$
\mathbb{E}\left[\left(\hat{V}_\kappa^\pi\right)^2\right] = \mathbb{E}\left[\left(\sum_{h=1}^H \left(\hat{r}_{\kappa,h}^\pi\right)\right)^2\right]
$$

$$
\leq H \sum_{h=1}^H \mathbb{E}\left[\left(\hat{r}_{\kappa,h}^\pi\right)^2\right] \leq H \sum_{h=1}^H \left\|\phi_{\pi,h}\right\|_{\tilde{\Sigma}_{\kappa,h}^{-1}}^2
$$

$$
\leq 2H \sum_{h=1}^H \left\|\phi_{\pi,h}\right\|_{\Sigma_{\kappa,h}^{-1}}^2 \leq 2H \left(\frac{\kappa \hat{\Delta}_\pi^2}{\beta_\kappa} + 4dH\right) .
$$

Using the properties of the Catoni estimator, we have:

$$
\left|\sum_{\kappa=k_0}^k V_\kappa^\pi - (k - k_0)\mathrm{Rob}_{k,\pi}\right|
$$

$$
\leq \alpha_k^\pi \sum_{\kappa=k_0+1}^k \left(\mathbb{E}\left[\left(\hat{V}_\kappa^\pi - V_\kappa^\pi\right)^2\right] + \left(V_\kappa^\pi - \frac{1}{k - k_0}\sum_{\kappa=k_0}^k V_\kappa^\pi\right)^2\right) + \frac{2\log\left(\frac{k^2|\Pi|}{\delta}\right)}{\alpha_k^\pi}
$$

$$
\leq \alpha_k^\pi \sum_{\kappa=k_0+1}^k H\left(2\frac{\kappa \hat{\Delta}_\pi^2}{\beta_\kappa} + 9dH\right) + \frac{4\log\left(\frac{k|\Pi|}{\delta}\right)}{\alpha_k^\pi} \tag{25}
$$

$$
\leq 4\sqrt{H \log\left(\frac{k|\Pi|}{\delta}\right) \sum_{\kappa=k_0+1}^k \left(2\frac{\kappa \hat{\Delta}_\pi^2}{\beta_\kappa} + 9dH\right)} . \tag{26}
$$

Where equation 26 is by our choice of $\alpha_k^\pi$.

Since equation 3 and equation 4 provides that:

$$
\hat{\Delta}_\pi = \frac{1}{k_0}\sum_{s=1}^{k_0} \hat{V}_s^{\hat{\pi}} - \hat{V}_s^\pi \geq 20\sqrt{\frac{f_K C_1}{k_0}} = 20\sqrt{\frac{f_K \beta_K dH^2}{k_0}} , \tag{27}
$$

we have $9dH \leq \frac{k_0 \hat{\Delta}_\pi^2}{\beta_K} \leq \frac{2\kappa \hat{\Delta}_\pi^2}{\beta_\kappa}$.

Thus we have:

$$
\left|\sum_{\kappa=k_0}^k V_k^\pi - (k - k_0)\mathrm{Rob}_{k,\pi}\right| \leq 4\sqrt{H \log\left(\frac{k|\Pi|}{\delta}\right) \sum_{\kappa=k_0+1}^k 4\frac{\kappa \hat{\Delta}_\pi^2}{\beta_\kappa}}
$$

$$
\leq 4\sqrt{H \log\left(\frac{k|\Pi|}{\delta}\right) \frac{4k}{\beta_k} \sum_{\kappa=k_0+1}^k \hat{\Delta}_\pi^2}
$$

$$
\leq \frac{1}{16}k\hat{\Delta}_\pi .
$$

Combining terms, we have:

$$
\left|\sum_{s=1}^k V_s^\pi - \sum_{s=1}^{k_0} \hat{V}_s^\pi - (k - k_0)\mathrm{Rob}_{k,\pi}\right| \leq \frac{1.4k\hat{\Delta}_\pi}{10} . \tag{28}
$$

Finally, we bound the deviation of $\hat{V}^{\hat{\pi}}$.

In the first $k_0$ episodes, since $\hat{\Delta}_{\hat{\pi}} = 0$, according to equation 24, we have that:

$$
\left|\sum_{s=1}^{k_0} V_s^{\hat{\pi}} - \hat{V}_s^{\hat{\pi}}\right| \leq \sqrt{f_k C_1 k_0} . \tag{29}
$$

In phase 2, since $\mathbb{E}\left[\hat{V}_k^{\hat{\pi}}\right] = V_k^{\hat{\pi}}$ is an unbiased estimator of the true value function, according to Freedman inequality, and that $\mathbb{E}\left[\left(\hat{V}_k^{\hat{\pi}}\right)^2\right] = \mathbb{E}\left[\frac{Y_k^2}{\tilde{p}_k^2(\hat{\pi})}\mathbb{1}\{\pi_k = \hat{\pi}\}\right] \le \frac{H^2}{\tilde{p}_k(\hat{\pi})} \le 2H^2$, we have:

$$\left|\sum_{s=k_0+1}^{k} \hat{V}_s^{\hat{\pi}} - V_s^{\hat{\pi}}\right| \le 2\sqrt{2H^2 k \log \frac{k|\Pi|}{\delta}} + 2H \log \frac{k|\Pi|}{\delta} \le \sqrt{C_1 k}\,.$$

Combining the two terms, we have:

$$\left|\sum_{s=1}^{k} \hat{V}_s^{\hat{\pi}} - V_s^{\hat{\pi}}\right| \le 2\sqrt{f_K C_1 k}\,. \tag{30}$$

In sum,

$$\sum_{s=1}^{k} V_s^{\hat{\pi}} - V_s^{\pi}$$

$$\ge \sum_{s=1}^{k-1} V_s^{\hat{\pi}} - V_s^{\pi} - 2H$$

$$\ge \sum_{s=1}^{k_0} \left(\hat{V}_s^{\hat{\pi}} - \hat{V}_s^{\pi}\right) + \left(\sum_{s=k_0+1}^{k-1} \hat{V}_s^{\hat{\pi}} - (k - k_0 - 1)\mathrm{Rob}_{k-1,\pi}\right) - 2\sqrt{f_K C_1(k-1)} - \frac{1.4(k-1)\hat{\Delta}_\pi}{10} - 2H$$

$$\ge (k-1)\hat{\Delta}_{k-1,\pi} - \frac{3(k-1)\hat{\Delta}_\pi}{10} \ge 0\,,$$

where the second to last inequality is due to equation 27, and the last one is due to equation 8.

$\square$

*Proof of Theorem 3.* Finally, we proof the regret bound in adversarial setting. For the regret in phase 2, we can decompose it as follows.

$$\sum_{s=k_0+1}^{k} V_s^{\hat{\pi}} - V_s^{\pi_s}$$

$$= \sum_{s=k_0+1}^{k} \left(\hat{V}_s^{\hat{\pi}} - Y_s\right) + \left(Y_s \mathbb{1}\{\pi_s = \hat{\pi}\} + V_s^{\hat{\pi}}\mathbb{1}\{\pi_s \ne \hat{\pi}\} - \hat{V}_s^{\hat{\pi}}\right)$$

$$+ \left[\left(V_s^{\hat{\pi}}\mathbb{1}\{\pi_s = \hat{\pi}\} - Y_s\mathbb{1}\{\pi_s = \hat{\pi}\}\right) + (Y_s - V_s^{\pi_s})\right]\,.$$

The first term is bounded by equation 9:

$$\sum_{s=k_0+1}^{k} \hat{V}_s^{\hat{\pi}} - Y_s \le \mathcal{O}\left(\sqrt{f_K C_1 k_0}\right)\,.$$

The second term is a martingale difference sequence since:

$$\mathbb{E}\left[Y_s \mathbb{1}\{\pi_s = \hat{\pi}\} + V_s^{\hat{\pi}}\mathbb{1}\{\pi_s \ne \hat{\pi}\} - \hat{V}_s^{\hat{\pi}}\right]$$

$$= \tilde{p}_s(\hat{\pi})\mathbb{E}\left[Y_s - \frac{Y_s}{\tilde{p}_s(\hat{\pi})}\right] + (1 - \tilde{p}_s(\hat{\pi})) V_s^{\hat{\pi}}$$

$$= \tilde{p}_s(\hat{\pi})\left(1 - \frac{1}{\tilde{p}_s(\hat{\pi})}\right)\mathbb{E}\left[Y_s\right] + (1 - \tilde{p}_s(\hat{\pi})) V_s^{\hat{\pi}}$$

$$= 0\,.$$

It's variance is bounded as:

$$\mathbb{E}\left[\left(Y_s \mathbb{1}\{\pi_s = \hat{\pi}\} + V_s^{\hat{\pi}} \mathbb{1}\{\pi_s \neq \hat{\pi}\} - \hat{V}_s^{\hat{\pi}}\right)^2\right]$$

$$= \tilde{p}_s(\hat{\pi}) \mathbb{E}\left[\left(Y_s - \frac{Y_s}{\tilde{p}_s(\hat{\pi})}\right)^2\right] + \left(1 - \tilde{p}_s(\hat{\pi})\right)\left(V_s^{\hat{\pi}}\right)^2$$

$$\leq \tilde{p}_s(\hat{\pi})\left(1 - \frac{1}{\tilde{p}_s(\hat{\pi})}\right)^2 H^2 + \left(1 - \tilde{p}_s(\hat{\pi})\right) H^2$$

$$\leq 2H^2\left(1 - \tilde{p}_s(\hat{\pi})\right),$$

where the last inequality is due to $\tilde{p}_s(\hat{\pi}) \geq \frac{1}{2}$.
The third term is also a martingale difference sequence, whose variance is bounded as:

$$\mathbb{E}\left[\left(\left(V_s^{\hat{\pi}} \mathbb{1}\{\pi_s = \hat{\pi}\} - Y_s \mathbb{1}\{\pi_s = \hat{\pi}\}\right) + \left(Y_s - V_s^{\pi_s}\right)\right)^2\right]$$

$$\leq \left(1 - \tilde{p}_s(\hat{\pi})\right) \mathbb{E}\left[\left(Y_s - V_s^{\pi_s}\right)^2\right]$$

$$\leq \left(1 - \tilde{p}_s(\hat{\pi})\right) 4H^2.$$

Also, we have:

$$1 - \tilde{p}_s(\hat{\pi}) \leq \frac{1}{2}p_s(\hat{\pi}) = \frac{1}{2}\sum_{\pi \neq \hat{\pi}} p_s(\pi) \leq \frac{1}{2}\sum_{\pi \neq \hat{\pi}} p_s(\pi) \frac{\hat{\Delta}_\pi}{\hat{\Delta}_{min}} \leq \frac{12dH\beta_s}{\hat{\Delta}_{min}^2 s}. \tag{31}$$

Thus, using Freedman inequality on the last two terms, we got:

$$\sum_{s=k_0+1}^{k} V_s^{\hat{\pi}} - V_s^{\pi_s}$$

$$\leq \mathcal{O}\left(\sqrt{f_K C_1 k_0} + \sqrt{\log\frac{k}{\delta}H^2 \sum_{s=1}^{k}\frac{dH\beta_s}{\hat{\Delta}_{min}^2 s}} + H\log\frac{k}{\delta}\right)$$

$$\leq \mathcal{O}\left(\sqrt{f_K C_1 k_0} + \sqrt{\frac{\log\left(\frac{k}{\delta}\right)dH^3 \log(k)\beta_k k_0}{f_K C_1}} + H\log\frac{k}{\delta}\right) \tag{32}$$

$$\leq \mathcal{O}\left(\sqrt{f_K C_1 k_0}\right).$$

Combining with the regret in phase 1, the regret in one epoch is bounded as:

$$\sum_{s=1}^{k} V_s^{\hat{\pi}} - V_s^{\pi_s} \leq \mathcal{O}\left(\sqrt{f_K C_1 k_0}\right).$$

Since the duration time $k_0$ of phase 1 is at least twice as the length of phase 1 in the previous epoch, we have that the sum of $\sqrt{k_0}$ in all the epochs is bounded by an constant factor of the square root of duration time in phase 1 of the last epoch, which is bounded by $\sqrt{K}$.
Summation over all the epochs, we have :

$$\text{Reg}(K; \Pi) = \mathcal{O}\left(\sqrt{C_1 K \log(K)}\right) = \mathcal{O}\left(\sqrt{dH^3 K \log(K)\log\left(\frac{|\Pi|K}{\delta}\right)}\right).$$

$\square$

---

**Algorithm 4** Geometric Hedge for Linear Adversarial MDP Policies

1: Input: policy set $\Pi$, $\gamma = \min\left\{\frac{1}{2}, \sqrt{\frac{dH\log\left(\frac{|\Pi|}{\delta}\right)}{K}}\right\}$, $\eta = \frac{\gamma}{4dH^2}$

2: Initialize: $\forall \pi \in \Pi, w_1(\pi) = 1, W_1 := |\Pi|$. $\forall h$ from 1 to $H$, compute the G-optimal design
   $g_h(\pi)$ on the set of feature visitations: $\{\phi_{\pi,h}, \pi \in \Pi\}$. Denote $g(\pi) = \frac{1}{H}\sum_{h=1}^{H} g_h(\pi)$

3: **for** each episode $k = 1$ to $K$ **do**

4:    $\forall \pi \in \Pi$,

$$p_k(\pi) = (1 - \gamma)\frac{w_k(\pi)}{W_k} + \gamma g(\pi)$$

5:    Select $\pi_k \in \Pi$ according to the probability $p_k(\pi)$ and collect rewards $y_{k,h}$

6:    Calculate reward estimators: $\hat{\theta}_{k,h} = \Sigma_{k,h}^{-1}\phi_{\pi_k,h}y_{k,h}$, and $\hat{r}_{k,h}^\pi = \phi_{\pi,h}^\top\hat{\theta}_{k,h}$, $\hat{V}_k^\pi = \sum_{h=1}^{H}\hat{r}_{k,h}^\pi$,
      where $\Sigma_{k,h} = \sum_\pi p_k(\pi)\phi_{\pi,h}\phi_{\pi,h}^\top$

7:    Compute the optimistic estimate of the value function:

$$\tilde{V}_k^\pi = \sum_{h=1}^{H}\left(\hat{r}_{k,h}^\pi + 2\phi_{\pi,h}^\top\Sigma_{k,h}^{-1}\phi_{\pi,h}\sqrt{H\frac{\log\left(\frac{1}{\delta}\right)}{dK}}\right)$$

8:    And we transform it into loss functions: $l_{k,h}(s_h, a_h) = 1 - r_{k,h}(s_h, a_h)$, $l_{k,h}^\pi = 1 - r_{k,h}^\pi$

$$L_{k,h}^\pi = \sum_{h=1}^{H} l_{k,h}^\pi = H - V_{k,h}^\pi$$

$$\hat{L}_{k,h}^\pi = \sum_{h=1}^{H}\hat{l}_{k,h}^\pi = \sum_{h=1}^{H}\left(1 - \hat{r}_{k,h}^\pi\right) = H - \hat{V}_{k,h}^\pi$$

$$\tilde{L}_{k,h}^\pi = H - \tilde{V}_{k,h}^\pi$$

9:    update using the loss estimators:

$$\forall \pi \in \Pi, w_{k+1}(\pi) = w_k(\pi)\exp\left(-\eta\tilde{L}_k^\pi\right), W_{k+1} = \sum_{\pi\in\Pi} w_{k+1}(\pi)$$

10: **end for**

---

## D HIGH PROBABILITY GUARANTEE FOR ADVERSARIAL LINEAR MDP

First, we bound the deviation between the estimated value and the true value of policy $\pi$.

**Lemma 9.** *Denote* $\text{DEV}_{K,\pi} = \left|\sum_{k=1}^{K}\hat{V}_k^\pi - V_k^\pi\right|$, *then we have:*

$$\text{DEV}_{K,\pi} = \left|\sum_{k=1}^{K}\hat{V}_k^\pi - V_k^\pi\right| = \left|\sum_{k=1}^{K}\hat{L}_k^\pi - L_k^\pi\right|$$

$$\leq \frac{1}{C_2}\sum_{k=1}^{K}\sum_{h=1}^{H}\|\phi_{\pi,h}\|_{\Sigma_{k,h}^{-1}}^2\sqrt{\frac{H\log\left(\frac{1}{\delta}\right)}{dK}} + C_2\sqrt{dKH\log\left(\frac{1}{\delta}\right)} + 2\left(\frac{dH^2}{\gamma} + H\right)\log\left(\frac{1}{\delta}\right) .$$

*Proof.* First, we show that $\hat{V}_k^\pi$ is an unbiased estimate of $V_k^\pi$:

$$\mathbb{E}\left[\hat{V}_k^\pi\right] = \sum_{h=1}^{H}\mathbb{E}\left[\hat{V}_{k,h}^\pi\right] = \sum_{h=1}^{H}\phi_{\pi,h}^\top\Sigma_{k,h}^{-1}\mathbb{E}\left[\phi_{\pi_k,h}y_{k,h}\right] ,$$

using the tower rule of expectation, we have:

$$\mathbb{E}\left[\phi_{\pi_k,h}y_{k,h}\right] = \mathbb{E}\left[\phi_{\pi_k,h}r_{k,h}^{\pi_k}\right] = \mathbb{E}\left[\phi_{\pi_k,h}\phi_{\pi_k,h}^\top\theta_{k,h}\right] = \Sigma_{k,h}\theta_{k,h}\,,$$

so

$$\mathbb{E}\left[\hat{V}_k^\pi\right] = \sum_{h=1}^{H}\phi_{\pi,h}^\top\theta_{k,h} = V_k^\pi\,.$$

Denote $\sigma^2 = \sum_{k=1}^{K}\mathrm{Var}\left(\hat{V}_k^\pi\right)$, then:

$$\sigma^2 \leq \sum_{k=1}^{K}\mathbb{E}\left[\left(\sum_{h=1}^{H}\phi_{\pi,h}^\top\Sigma_{k,h}^{-1}\phi_{\pi_k,h}y_{k,h}\right)^2\right]$$

$$\leq \sum_{k=1}^{K}H\sum_{h=1}^{H}\mathbb{E}\left[\left(\phi_{\pi,h}^\top\Sigma_{k,h}^{-1}\phi_{\pi_k,h}y_{k,h}\right)^2\right]$$

$$\leq H\sum_{k=1}^{K}\sum_{h=1}^{H}\left\|\phi_{\pi,h}\right\|_{\Sigma_{k,h}^{-1}}^2\,.$$

Also, due to the properties of G-optimal design, we have:

$$\left\|\phi_{\pi,h}\right\|_{\left(\sum_\pi g_h(\pi)\phi_{\pi,h}\phi_{\pi,h}^\top\right)^{-1}}^2 \leq d\,,$$

and $\Sigma_{k,h} \succeq \frac{\gamma}{H}\sum_\pi g_h(\pi)\phi_{\pi,h}\phi_{\pi,h}^\top$. Thus we have $\left\|\phi_{\pi,h}\right\|_{\Sigma_{k,h}^{-1}}^2 \leq \frac{dH}{\gamma}, \forall \pi \in \Pi$, so $\hat{V}_k^\pi \leq \frac{dH^2}{\gamma}$. Using Freedman inequality, the sum of the martingale difference sequence $\hat{V}_k^\pi - V_k^\pi$ is bounded as:

$$\left|\sum_{k=1}^{K}\hat{V}_k^\pi - V_k^\pi\right| \leq 2\sqrt{H\sum_{k=1}^{K}\sum_{h=1}^{H}\left\|\phi_{\pi,h}\right\|_{\Sigma_{k,h}^{-1}}^2\log\left(\frac{1}{\delta}\right) + \left(\frac{dH^2}{\gamma}+H\right)\log\left(\frac{1}{\delta}\right)}$$

$$\leq \frac{1}{C_2}\sum_{k=1}^{K}\sum_{h=1}^{H}\left\|\phi_{\pi,h}\right\|_{\Sigma_{k,h}^{-1}}^2\sqrt{\frac{H\log\left(\frac{1}{\delta}\right)}{dK}} + C_2\sqrt{dKH\log\left(\frac{1}{\delta}\right)} + 2\left(\frac{dH^2}{\gamma}+H\right)\log\left(\frac{1}{\delta}\right)\,,$$

where the last inequality is due to the geometric mean-arithmetic mean inequality. $\qquad\square$

**Lemma 10.**

$$L_k^{\pi_k} - \sum_{k=1}^{K}\sum_\pi p_k(\pi)\hat{L}_k^\pi$$

$$= \sum_{k=1}^{K}\sum_\pi p_k(\pi)\hat{V}_k^\pi - \sum_{k=1}^{K}V_k^{\pi_k}$$

$$\leq H\left(\sqrt{d}+1\right)\sqrt{2K\log\left(\frac{1}{\delta}\right)} + \frac{4}{3}\left(H+\frac{dH^2}{\gamma}\right)\log\left(\frac{1}{\delta}\right)\,.$$

*Proof.*

$$\sum_{k=1}^{K}\sum_\pi p_k(\pi)\hat{V}_k^\pi - \sum_{k=1}^{K}V_k^{\pi_k} = \sum_{k=1}^{K}\sum_{h=1}^{H}\left(\sum_\pi p_k(\pi)\hat{r}_{k,h}^\pi - r_{k,h}^{\pi_k}\right)\,. \qquad (33)$$

Using lemma 6 in Bartlett et al. (2008), we have:

$$\left|\sum_{k=1}^{K}r_{k,h}^{\pi_k} - \sum_{k=1}^{K}\sum_\pi p_k(\pi)\hat{r}_{k,h}^\pi\right| \leq \left(\sqrt{d}+1\right)\sqrt{2K\log\left(\frac{1}{\delta}\right)} + \frac{4}{3}\left(\frac{dH}{\gamma}+1\right)\log\left(\frac{1}{\delta}\right)\,.$$

Since $\left|\hat{\theta}_{k,h}^\top\phi_{\pi,h}\right| \leq \frac{dH}{\gamma}$.

Plugging it into equation 33, we finish the proof. $\qquad\square$

**Lemma 11.** *With probability at least $1 - \delta$, we have :*

$$\sum_{k=1}^{K} \sum_{\pi} p_k(\pi) \left(\tilde{L}_k^{\pi}\right)^2 \leq 2(d+1)KH^2 + 2\frac{dH^3}{\gamma}\sqrt{2K \log\left(\frac{1}{\delta}\right)} + \frac{8dH^3 \log\left(\frac{1}{\delta}\right)}{\gamma}.$$

*Proof.* Since $\hat{L}_k^{\pi} = H - \hat{V}_k^{\pi}$, $\left(\hat{L}_k^{\pi}\right)^2 \leq \left(\hat{V}_k^{\pi}\right)^2 + H^2$, we have:

$$\sum_{k=1}^{K} \sum_{\pi} p_k(\pi) \left(\hat{L}_k^{\pi}\right)^2 \leq \sum_{k=1}^{K} \sum_{\pi} p_k(\pi) \left(\hat{V}_k^{\pi}\right)^2 + KH^2.$$

Using Cauchy-Schwarz inequality, we have:

$$\left(\hat{V}_k^{\pi}\right)^2 \leq H \sum_{h=1}^{H} \left(\hat{r}_{k,h}^{\pi}\right)^2.$$

So,

$$\sum_{k=1}^{K} \sum_{\pi} p_t(\pi) \left(\hat{r}_{k,h}^{\pi}\right)^2$$
$$= \sum_{k=1}^{K} \sum_{\pi} p_t(\pi) \hat{\theta}_{k,h}^{\top} \phi_{\pi,h} \phi_{\pi,h}^{\top} \hat{\theta}_{k,h}$$
$$= \sum_{k=1}^{K} \sum_{\pi} \hat{\theta}_{k,h}^{\top} \Sigma_{k,h} \hat{\theta}_{k,h}$$
$$\leq \sum_{k=1}^{K} \phi_{\pi_k,h}^{\top} \Sigma_{k,h}^{-1} \phi_{\pi_k,h}.$$

Notice that using the definition of $\Sigma_{k,h}$ and properties of the G-optimal design,

$$\mathbb{E}\left[\phi_{\pi_k,h}^{\top} \Sigma_{k,h}^{-1} \phi_{\pi_k,h}\right] = d,$$

$$\phi_{\pi_k,h}^{\top} \Sigma_{k,h}^{-1} \phi_{\pi_k,h} \leq \frac{dH}{\gamma}.$$

Applying the Hoeffding bound, we got:

$$\sum_{k=1}^{K} \sum_{\pi} p_t(\pi) \left(\hat{r}_{k,h}^{\pi}\right)^2 \leq dK + \frac{dH}{\gamma}\sqrt{2K \log\left(\frac{1}{\delta}\right)}.$$

Thus:

$$\sum_{k=1}^{K} \sum_{\pi} p_k(\pi) \left(\hat{V}_k^{\pi}\right)^2$$
$$\leq H \sum_{h=1}^{H} \sum_{k=1}^{K} \sum_{\pi} p_t(\pi) \left(\hat{r}_{k,h}^{\pi}\right)^2$$
$$\leq dKH^2 + \frac{dH^3}{\gamma}\sqrt{2K \log\left(\frac{1}{\delta}\right)}.$$

And:

$$
\left(\tilde{L}_k^\pi\right)^2 = \left(\hat{L}_k^\pi - \sum_{h=1}^H 2\phi_{\pi,h}^\top \Sigma_{k,h}^{-1}\phi_{\pi,h}\sqrt{H\frac{\log\left(\frac{1}{\delta}\right)}{dK}}\right)^2
$$

$$
\leq 2\left(\hat{L}_k^\pi\right)^2 + 2\left(\sum_{h=1}^H 2\phi_{\pi,h}^\top \Sigma_{k,h}^{-1}\phi_{\pi,h}\sqrt{H\frac{\log\left(\frac{1}{\delta}\right)}{dK}}\right)^2
$$

$$
\leq 2\left(\hat{L}_k^\pi\right)^2 + 2H\sum_{h=1}^H 4\left\|\phi_{\pi,h}\right\|_{\Sigma_{k,h}^{-1}}^2 \phi_{\pi,h}^\top \Sigma_{k,h}^{-1}\phi_{\pi,h}\frac{H\log\left(\frac{1}{\delta}\right)}{dK}
$$

$$
\leq 2\left(\hat{V}_k^\pi\right)^2 + 2H^2 + 8\frac{dH}{\gamma}\sum_{h=1}^H \phi_{\pi,h}^\top \Sigma_{k,h}^{-1}\phi_{\pi,h}\frac{H\log\left(\frac{1}{\delta}\right)}{dK}\ .
$$

So we conclude that:

$$
\sum_{k=1}^K \sum_\pi p_k(\pi)\left(\tilde{L}_k^\pi\right)^2
$$

$$
\leq 2\sum_{k=1}^K \sum_\pi p_k(\pi)\left(\hat{L}_k^\pi\right)^2 + \frac{8H^2\log\frac{1}{\delta}}{\gamma K}\sum_{k=1}^K \sum_{h=1}^H \sum_\pi p_k(\pi)\phi_{\pi,h}^\top \Sigma_{k,h}^{-1}\phi_{\pi,h}
$$

$$
\leq 2(d+1)KH^2 + 2\frac{dH^3}{\gamma}\sqrt{2K\log\left(\frac{1}{\delta}\right)} + \frac{8dH^3\log\left(\frac{1}{\delta}\right)}{\gamma}\ .
$$

$\square$

*Proof of Theorem 1 .* Now we analyze the potential function. Using classical techniques, we got that counter part of equation (2) in Bartlett et al. (2008):

$$
\log\left(\frac{W_K}{W_1}\right) = \sum_{k=1}^K \log\left(\frac{W_k}{W_{k-1}}\right)
$$

$$
= \sum_{k=1}^K \log\left(\sum_\pi \frac{w_k(\pi)}{W_{k-1}}\exp\left(-\eta\tilde{L}_k^\pi\right)\right)
$$

$$
\leq \sum_{k=1}^K \log\left(\sum_\pi \frac{p_k(\pi) - \gamma g_\pi}{1-\gamma}\left(1 - \eta\tilde{L}_k^\pi + \eta^2\left(\tilde{L}_k^\pi\right)^2\right)\right) \tag{34}
$$

$$
\leq \sum_{k=1}^K \sum_\pi \frac{p_k(\pi) - \gamma g_\pi}{1-\gamma}\left(-\eta\tilde{L}_k^\pi + \eta^2\left(\tilde{L}_k^\pi\right)^2\right)
$$

$$
\leq \frac{\eta}{1-\gamma}\left[\sum_{k=1}^K \sum_\pi -p_k(\pi)\tilde{L}_k^\pi + \gamma\sum_{k=1}^K \sum_\pi g(\pi)\tilde{L}_k^\pi + \eta\sum_{k=1}^K \sum_\pi p_k(\pi)\left(\tilde{L}_k^\pi\right)^2\right]\ , \tag{35}
$$

where inequality 34 is due to the constraint that $\left|\eta \tilde{L}_k^\pi\right| \le 1$.

Using Lemma 10, we have:

$$\sum_{k=1}^{K} \sum_\pi \sum_\pi -p_k(\pi)\tilde{L}_k^\pi$$

$$=\sum_{k=1}^{K} \sum_\pi -p_k(\pi)\hat{L}_k^\pi + 2\sum_{k=1}^{K}\sum_{h=1}^{H}\sum_\pi p_k(\pi)\phi_{\pi,h}^\top \Sigma_{k,h}^{-1}\phi_{\pi,h}\sqrt{H\frac{\log\left(\frac{1}{\delta}\right)}{dK}}$$

$$=\sum_{k=1}^{K} \sum_\pi -p_k(\pi)\hat{L}_k^\pi + 2H\sqrt{dKH\log\left(\frac{1}{\delta}\right)}$$

$$\le \sum_{k=1}^{K} -L_k^{\pi_k} + H\left(\sqrt{d}+1\right)\sqrt{2K\log\left(\frac{1}{\delta}\right)} + \frac{4}{3}\left(H+\frac{dH^2}{\gamma}\right)\log\left(\frac{1}{\delta}\right) + 2H\sqrt{dKH\log\left(\frac{1}{\delta}\right)}.$$

Using Lemma 9 and choosing $C_2 = \frac{1}{2}$, we have:

$$\sum_{k=1}^{K}\tilde{L}_k^\pi \le \sum_{k=1}^{K} L_k^\pi + \mathrm{DEV}_{k,\pi} - \sum_{k=1}^{K}\sum_{h=1}^{H} 2\phi_{\pi,h}^\top\Sigma_{k,h}^{-1}\phi_{\pi,h}\sqrt{H\frac{\log\left(\frac{1}{\delta}\right)}{dK}}$$

$$\le \sum_{k=1}^{K} L_k^\pi + \frac{1}{2}\sqrt{dKH\log\left(\frac{1}{\delta}\right)} + 2\left(\frac{dH^2}{\gamma}+H\right)\log\left(\frac{1}{\delta}\right)$$

$$\le KH + \frac{1}{2}\sqrt{dKH\log\left(\frac{1}{\delta}\right)} + 2\left(\frac{dH^2}{\gamma}+H\right)\log\left(\frac{1}{\delta}\right).$$

Thus equation 35 becomes:

$$\log\left(\frac{W_K}{W_1}\right) \le \frac{\eta}{1-\gamma}\left[-\sum_{k=1}^{K} L_k^{\pi_k} + H\left(\sqrt{d}+1\right)\sqrt{2K\log\left(\frac{1}{\delta}\right)} + \frac{4}{3}\left(H+\frac{dH^2}{\gamma}\right)\log\left(\frac{1}{\delta}\right) + 2H\sqrt{dKH\log\left(\frac{1}{\delta}\right)}\right]$$

$$+ 2\eta\gamma KH + \eta\left[\frac{1}{2}\sqrt{dKH\log\left(\frac{1}{\delta}\right)} + 2\left(\frac{dH^2}{\gamma}+H\right)\log\left(\frac{1}{\delta}\right)\right]$$

$$+ 2\eta^2\left[2(d+1)KH^2 + 2\frac{dH^3}{\gamma}\sqrt{2K\log\left(\frac{1}{\delta}\right)} + \frac{8dH^3\log\left(\frac{1}{\delta}\right)}{\gamma}\right]$$

$$\le \eta\left(-\sum_{k=1}^{K} L_k^{\pi_k}\right) + 2\eta^2\left[2(d+1)KH^2 + 2\frac{dH^3}{\gamma}\sqrt{2K\log\left(\frac{1}{\delta}\right)} + \frac{8dH^3\log\left(\frac{1}{\delta}\right)}{\gamma}\right]$$

$$+ 2\eta\left[H\left(\sqrt{d}+1\right)\sqrt{2K\log\left(\frac{1}{\delta}\right)} + \frac{4}{3}\left(H+\frac{dH^2}{\gamma}\right)\log\left(\frac{1}{\delta}\right) + 2H\sqrt{dKH\log\left(\frac{1}{\delta}\right)}\right]$$

$$+ 2\eta\gamma KH + \eta\left[\frac{1}{2}\sqrt{dKH\log\left(\frac{1}{\delta}\right)} + 2\left(\frac{dH^2}{\gamma}+H\right)\log\left(\frac{1}{\delta}\right)\right].$$

$$(36)$$

And we also have for $\forall \pi \in \Pi$,

$$
\begin{aligned}
\log\left(\frac{W_K}{W_1}\right) \geq & \eta\left(\sum_{k=1}^{K} -\tilde{L}_k^\pi\right) - \log\left(|\Pi|\right) \\
\geq & \eta\left(\sum_{k=1}^{K} -L_k^\pi - \mathrm{DEV}_{K,\pi} + \sum_{k=1}^{K}\sum_{h=1}^{H} 2\phi_{\pi,h}^\top \Sigma_{k,h}^{-1} \phi_{\pi,h}\sqrt{H\frac{\log\left(\frac{1}{\delta}\right)}{dK}}\right) - \log\left(|\Pi|\right) ,
\end{aligned}
$$

(37)

where the last inequality is due to Lemma 9.
Plugging equation 36 and equation 37 together, we have that for $\forall \pi \in \Pi$:

$$
\begin{aligned}
\sum_{k=1}^{K} L_k^{\pi_k} - L_k^\pi \leq & \mathrm{DEV}_{k,\pi} - \sum_{k=1}^{K}\sum_{h=1}^{H} 2\phi_{\pi,h}^\top \Sigma_{k,h}^{-1} \phi_{\pi,h}\sqrt{H\frac{\log\left(\frac{1}{\delta}\right)}{dK}} \\
& + 4\left(\frac{dH^2}{\gamma} + H\right)\log\left(\frac{1}{\delta}\right) + \frac{log|\Pi|}{\eta} + 5H\sqrt{dKH\log\left(\frac{1}{\delta}\right)} \\
& + 2H\left(\sqrt{d} + 1\right)\sqrt{2K\log\left(\frac{1}{\delta}\right)} + \frac{8}{3}\left(H + \frac{dH^2}{\gamma}\right)\log\left(\frac{1}{\delta}\right) \\
& + 2\gamma KH + 4\eta\left((d+1)KH^2 + \frac{dH^3}{\gamma}\sqrt{2K\log\left(\frac{1}{\delta}\right)}\right) + \frac{16\eta dH^3\log\frac{1}{\delta}}{\gamma} .
\end{aligned}
$$

(38)

When $K \geq L_0 = 4dH\log\left(\frac{|\Pi|}{\delta}\right)$ and $\gamma \leq \frac{1}{2}$,

$$
\left|\tilde{L}_k^\pi\right| \leq H + \left|\tilde{V}_k^\pi\right| \leq H + \frac{dH^2}{\gamma}\left(1 + 2\sqrt{\frac{H\log\left(\frac{1}{\delta}\right)}{dK}}\right) \leq \frac{4dH^2}{\gamma} .
$$

So the choice of $\eta = \frac{\gamma}{4dH^2}$ ensures $\left|\eta\tilde{L}_k^\pi\right| \leq 1$. Plugging in our choice of $\eta$, equation 38 then becomes:

$$
\begin{aligned}
\sum_{k=1}^{K} L_k^{\pi_k} - L_k^\pi \leq & \left(\frac{1}{C_2} - 2\right)\left(\sum_{k=1}^{K}\sum_{h=1}^{H} 2\phi_{\pi,h}^\top \Sigma_{k,h}^{-1} \phi_{\pi,h}\sqrt{H\frac{\log\left(\frac{1}{\delta}\right)}{dK}}\right) \\
& + \mathcal{O}\left(\sqrt{dH^3 K\log\left(\frac{1}{\delta}\right)} + \frac{dH^2}{\gamma}\log\left(\frac{|\Pi|}{\delta}\right) + \gamma KH\right) .
\end{aligned}
$$

Choosing $C_2 = \frac{1}{2}$, we have:

$$
\sum_{k=1}^{K} L_k^{\pi_k} - L_k^\pi \leq \mathcal{O}\left(\sqrt{dH^3 K\log\left(\frac{1}{\delta}\right)} + \frac{dH^2}{\gamma}\log\left(\frac{|\Pi|}{\delta}\right) + \gamma KH\right) .
$$

(39)

And by our choice of $\gamma = \min\left\{\frac{1}{2}, \sqrt{\frac{dH\log\left(\frac{|\Pi|}{\delta}\right)}{K}}\right\} = \sqrt{\frac{dH\log\left(\frac{|\Pi|}{\delta}\right)}{K}}$, we have:

$$
\mathrm{Reg}\left(K;\Pi\right) \leq \sum_{k=1}^{K} L_k^{\pi_k} - L_k^\pi \leq \mathcal{O}\left(\sqrt{dH^3 K\log\left(\frac{|\Pi|}{\delta}\right)}\right) .
$$

(40)

For $K \le L_0 = 4dH \log\left(\frac{|\Pi|}{\delta}\right)$, we have:

$$\mathrm{Reg}\left(K;\Pi\right) \le KH \le \sqrt{KL_0 H} = \mathcal{O}\left(\sqrt{dH^3 K \log\left(\frac{|\Pi|}{\delta}\right)}\right). \tag{41}$$

Combining equation 41 and equation 40, we prove the regret bound.

Moreover, when $K \ge L_0$, choosing $C_2 \ge 20$ and recalling the definition of $\mathrm{DEV}_{k,\pi}$ in Lemma 9,

$$
\begin{aligned}
\sum_{k=1}^{K} V_k^{\pi} - V_k^{\pi_k} &= \sum_{k=1}^{K} L_k^{\pi_k} - L_k^{\pi} \\
&\le -\sum_{k=1}^{K}\sum_{h=1}^{H} 2\phi_{\pi,h}^{\top} \Sigma_{k,h}^{-1} \phi_{\pi,h} \sqrt{H \frac{\log\left(\frac{1}{\delta}\right)}{dK}} \\
&\quad + \mathcal{O}\left(\sqrt{dH^3 K \log\left(\frac{1}{\delta}\right)} + \frac{dH^2}{\gamma} \log\left(\frac{|\Pi|}{\delta}\right) + \gamma KH\right) \\
&\le -C_2 \mathrm{DEV}_{K,\pi} + \mathcal{O}\left(\sqrt{dH^3 K \log\left(\frac{1}{\delta}\right)} + \frac{dH^2}{\gamma} \log\left(\frac{|\Pi|}{\delta}\right) + \gamma KH\right).
\end{aligned}
$$

Applying a union bound for all the possible $k_0 \in \{1,2,\cdots K\}$, we conclude with at least probability $1 - \delta$, we have:

$$\sum_{k=1}^{k_0} V_k^{\pi} - V_k^{\pi_k} \le \sqrt{C_1 k_0} - C_2 \mathrm{DEV}_{k_0,\pi}.$$

With the constant $C_1 = \mathcal{O}\left(dH^3 \log\left(\frac{|\Pi| K}{\delta}\right)\right) \ge dH^2 \beta_K$, proving Theorem 1.

$\square$

## E    CONCENTRATION INEQUALITIES

**Lemma 12.** *[Freedman inequality(Freedman, 1975)] Let $\mathcal{F}_0 \subset \mathcal{F}_1 \subset \cdots \subset \mathcal{F}_T$ be a filtration and let $X_1, X_2, \cdots X_T$ be random variables such that $X_t$ is $\mathcal{F}_t$ measurable, $\mathbb{E}\left[X_t | \mathcal{F}_{t-1}\right] = 0$, $|X_t| \le b$ almost surely, and $\sum_{t=1}^{T} \mathbb{E}\left[X_t^2 | \mathcal{F}_{t-1}\right] \le V$ for some fixed $V > 0$ and $b > 0$. Then, for any $\delta \in (0,1)$, we have with probability at least $1 - \delta$,*

$$\sum_{t=1}^{T} X_t \le 2\sqrt{V \log\left(1/\delta\right)} + b \log\left(1/\delta\right).$$

**Lemma 13** (Concentration inequality for Catoni estimators (Wei & Luo, 2018; Lee et al., 2021)). *Let $\mathcal{F}_0 \subset \mathcal{F}_1 \subset \cdots \subset \mathcal{F}_n$ be a filtration and let $X_1, X_2, \cdots X_n$ be random variables such that $X_i$ is $\mathcal{F}_i$ measurable, $\mathbb{E}\left[X_i | \mathcal{F}_{i-1}\right] = \mu_i$ for some fixed $\mu_i$, and $\sum_{i=1}^{n} \mathbb{E}\left[(X_i - \mu_i)^2 | \mathcal{F}_{i-1}\right] \le V$ for some fixed $V$. Denote $\mu = \frac{1}{n}\sum_{i=1}^{n} \mu_i$ and let $\hat{\mu}_{n,\alpha}$ be the Catoni's robust mean estimator of $X_1, X_2, \cdots X_n$ with a fixed parameter $\alpha$, that is, $\hat{\mu}_{n,\alpha}$ is the unique root of the function: $f(z) = \sum_{i=1}^{n} \Phi\left(\alpha\left(X_i - z\right)\right)$, where $\Phi(y) = \log\left(1 + y + y^2/2\right)$ if $y \ge 0$ and $\Phi(y) = -\log\left(1 - y + y^2/2\right)$ otherwise.*

*Then for any $\delta \in (0,1)$, as long as $n$ is large enough that $n \ge \alpha^2\left(V + \sum_{i=1}^{n}\left(\mu_i - \mu\right)^2\right) + 2\log\left(1/\delta\right)$, we have with probability at least $1 - 2\delta$,*

$$\left|\hat{\mu}_{n,\alpha} - \mu\right| \le \frac{\alpha\left(V + \sum_{i=1}^{n}\left(\mu_i - \mu\right)^2\right)}{n} + \frac{2\log\left(1/\delta\right)}{\alpha n}.$$

*Choosing $\alpha$ optimally, we have:*

$$\left| \hat{\mu}_{n,\alpha} - \mu \right| \leq \frac{2}{n} \sqrt{2 \left( V + \sum_{i=1}^{n} (\mu_i - \mu)^2 \right) \log \left( 1/\delta \right)}.$$

*In particular, if $\mu_1 = \mu_2 = \cdots = \mu_n$, we have:*

$$\left| \hat{\mu}_{n,\alpha} - \mu \right| \leq \frac{2}{n} \sqrt{2V \log \left( 1/\delta \right)}.$$

