# OpenReview forum: "Simultaneously Learning Stochastic and Adversarial Markov Decision Process with Linear Function Approximation"
_ICLR.cc/2023/Conference — Submitted to ICLR 2023_

### Official Review · Reviewer_PB4v · 2022-10-20

**Confidence:** 2
**Correctness:** 3
**Technical Novelty And Significance:** 2
**Empirical Novelty And Significance:** Not applicable
**Recommendation:** 5

**Clarity, Quality, Novelty And Reproducibility:**

I am not particularly well-read on this particular line of work, and am taking the authors at their word that this problem has not been tackled in the past. The problem tackled in the paper is quite natural to state, and seems to be of obvious interest in this area. However, as discussed above, I am concerned that the results of the paper seem to **a)** follow pretty immediately from previous results in the non-MDP bandit setting, namely those of Lee et al (2021) and **b)** not really use anything particular about MDPs. As such, I am not sold on the novelty of this paper's approach, or results, compared to Lee et al (2021).

The paper is clearly written, and there are no reproducibility concerns--there are no experiments (the paper is completely theoretical), and all proofs are in the appendix.

**Strength And Weaknesses:**

The authors tackle an interesting problem and derive a new algorithm that achieves a "best-of-both-worlds" guarantee. The paper is clearly written, and the problem tackled is interesting. However, I do have some technical concerns.

1. I would like some elaboration on the covering approach used to move from finite hypothesis class $\Pi$ to infinite hypothesis class. Let's say, for example, that I want my hypothesis class to be the set of policies $\pi_h(s) = \text{argmax}_a \langle \phi(s, a), w_h) \rangle$ for some weights $w_h$---that is, $W = (w_1, \dots, w_H) \in \mathbb{R}^{d \times H}$ parameterizes my Q estimates. Naively, I would try to $\epsilon$-cover the space of all such $W$ by considering discretizations of (say) $[0, 1]^{d\times H}$ to precision $\epsilon$. We would thus have $|\Pi| = (1/\epsilon)^{dH}$.

    **a)** This would lose factors of $\sqrt{dH}$ and $dH$ in the adversarial and stochastic regret bounds respectively. I feel like that means it is worth making explicit in the paper--it *does* "influence the main order of the regret".

    **b)** If we use such a cover by exponentially many functions, is the algorithm stated in the paper efficient? Many steps in the algorithm seem to involve enumerating policies $\pi\in\Pi$, which would take exponential time.

1. The factor $\Delta_\min$ used in this paper seems quite unusually small. For example, Jin & Luo (2020) are able to achieve a bound in the tabular setting with $\Delta_\min := \min_{h, (s, a):a \ne \pi^*(s)} V^*_h(s) -  Q_h^*(s, a)$ (which is larger, at least if the hypothesis class $\Pi$ consists of all deterministic policies). Is it possible for this paper to match that $\Delta_\min$?

1. The algorithm seems broadly extremely similar to Lee et al (2021)'s algorithm for the (non-MDP) bandit setting, applied to the set of arms $\mathcal{X} = \\{ \phi_{\pi, \cdot} : \pi \in \Pi \\}$. What, if any, is the difference here?

More generally, it does not seem as though this paper is really using any feature of MDPs--instead, it seems that it is directly applying a non-MDP result to the MDP setting and simply ignoring most of the structure of the MDP. As a result, the main theorems of this paper do not look like what I would typically expect MDP results to look like--for example, I would expect MDP results to have dependencies on "local" properties such as the larger $\Delta_\min$ (see point 2 above) and "local" hypothesis classes, rather than the "global" $\Delta_\min$ and "global" policies $\pi \in \Pi$.

Minor detail: at various points throughout the paper, $O(\sqrt{K})$ should be $\widetilde O(\sqrt{K})$ since the adversarial regret has $\log K$ dependencies.

**Summary Of The Paper:**

The authors tackle the setting of linear MDPs with known transition but unknown reward that achieves $\text{polylog}(K)$ regret over $K$ episodes when the rewards are stochastic, and $\tilde O(\sqrt{K})$ regret when the rewards are adversarial.

**Summary Of The Review:**

A theoretical paper that tackles an interesting and clearly important problem. However, I am not convinced that it tackles the problem sufficiently to distinguish itself from past works. I am willing to consider raising my score if the reviewers provide a response that alleviates my concerns.

---

> ### Author Response · Authors · 2022-11-18
> **Reply to Reviewer PB4v**
>
> We thank the reviewer for the valuable suggestions and comments. Please find our reply below.
>
> -covering based on greedy policies
>
> Thanks for your suggestion. Yes, such a covering can ensure that the value of the optimal policy can be approximated by a representative in the covering. But there are some other issues that need to be considered to apply our theoretical analysis. One is that there may be more than one optimal policy in the covering, which violates the uniqueness assumption. The other is that the minimum sub-optimality gap in such a policy set would be epsilon, which would be small when we want to align the regret definition with He et al. (2021) by setting epsilon to be $1/T$. So such a covering may not be appropriate for the current analysis. In the next question, we discuss the difference between our gap and that of He et al. (2021).
>
> For the computation problem, yes, the computational complexity depends on the size of the policy set. And it may be high when the number of policies becomes exponentially large. Our approach can be regarded as an attempt at the small policy set case. We think it is also natural to first give possible solutions under a simpler case. For example, there are also many attempts at linear MDP in the stochastic setting before the first efficient LSVI-UCB algorithm was proposed/proved by Jin et al. (2020b).
>
> -difference between our $\Delta$ and that in LSVI-UCB
>
> Fix a policy $\pi$, to compare our $\Delta_{\pi}$ with $gap_\min$ in He at al. (2021), we assume the policy set $\Pi$ contains the real optimal policy $\pi^*$.
> In general, $\Delta_\pi = \mathbb{E}\left[\sum_{h=1}^H gap_h(s_h,a_h) \mid \pi \right] = \sum_{h,s,a} p^\pi_h(s,a) gap_h(s,a)$ where the first equality follows Eq (B.2) in He et al. (2021) and $p^\pi_h(s,a)$ is the visitation probability of state-action $(s,a)$ at step $h$ by following $\pi$. This shows that $\Delta_\pi \ge p^\pi gap_\min$ in the worst case where $p^{\pi}$ is the minimum none-zero visitation probability of policy $\pi$ at some state-action pair.
>
> And there are also cases where our defined $\Delta_{\pi}$ is larger than that in He et al. (2021). When both the policy and transition are deterministic (some references on deterministic transition are listed below), we have
> $\Delta_\pi = \sum_{h=1}^H gap_h(s_h,a_h) \ge gap_\min$.
> And in the stochastic transition case, if all sub-optimal policies happen to not select the optimal action in $\arg\max_{a} Q_1^*(s_1,a)$,
> $\Delta_{\pi}= V_1^*(s_1)-V_1^{\pi}(s_1) =  V_1^*(s_1)-Q_1^{\pi}(s_1,a') \ge V_1^*(s_1)-Q_1^{*}(s_1,a') =\text{gap}_1(s_1,a').$
> where $a'$ is the action selected by $\pi$ at the first step and $ \text{gap}_1(s_1,a') >0$ since $a'$ is not in the optimal action set at $s_1$. In the above two cases, our sub-optimality gap is larger than the previously defined gap and our dependence on the gap is better. We also add this discussion in the revision.
>
> Andrea Tirinzoni, Aymen Al-Marjani and Emilie Kaufmann. Near Instance-Optimal PAC Reinforcement Learning for Deterministic MDPs. NeurIPS 2022.
>
> Chris Dann et al. Beyond Value-Function Gaps: Improved Instance-Dependent Regret Bounds for Episodic Reinforcement Learning. NeurIPS 2021.
>
> Ronald Ortner. Online regret bounds for Markov decision processes with deterministic transitions. Theoretical Computer Science. 2010.
> Damianos Tranos and Alexandre Proutiere. Regret Analysis in Deterministic Reinforcement Learning. 60th IEEE Conference on Decision and Control (CDC). IEEE, 2021.
>
>
> -novelty, non-MDP structure
>
> Our approach relies on the novel observation that the value of a policy can be written as the inner product of the visitation feature and an unknown reward parameter. From this view, we are able to reduce the problem to linear optimization and existing techniques for linear optimization can be used. It is worth noting that reducing a hard problem to a well-understood setting is a typical way to solve the problem and has its own contribution. For example, the widely studied occupancy measure-based approach also reduces harder MDP to the linear optimization problem and aims to optimize the policy by optimizing the occupancy measure. Though we regard a policy as a whole, the problem hardness $\Delta$ is related to the previously defined local gap. We have to emphasize that we are the first to introduce this type of linear optimization for the regret minimization problem in MDP and we believe such reduction may be of independent interest.
>
> -minor: $O(K)->\tilde{O}(K)$
>
> We thank the reviewer for the detailed comment. We modified it in the revision.

---

> > ### Comment · Reviewer_PB4v · 2022-11-18
> > **Response**
> >
> > Thank you for your response. It has not significantly changed my view of the paper, and if anything it has confirmed my primary concern. Namely, the paper's main technical contribution seems not to be the new algorithm or its analysis, but rather the definition of $\phi_\pi$, after which the main theorem follows directly from a black-box application of the main result of Lee et al. Also, the definition of $\phi_\pi$ is, in my opinion, not significantly novel either: it's just the expected feature vector, which, as the authors point out, satisfies the required linearity conditions as a straightforward consequence of the linearity of expectation.
> > As such, at a minimum the paper seems improperly positioned--the nature of the contribution, and its relation especially to Lee et al, should be made clearer.
> >
> > Regarding the time complexity of the proposed method, I believe that in a future revision this weakness should be made clear and discussed---in general MDPs it seems like a very strong assumption to assume a polynomial-sized policy set, and with an exponential-sized policy set the algorithm basically is only of interest in providing an information-theoretic guarantee, as the size of the policy set would make the actual executing of the algorithm impossible.

---

### Official Review · Reviewer_WXEg · 2022-10-23

**Confidence:** 4
**Correctness:** 4
**Technical Novelty And Significance:** 3
**Empirical Novelty And Significance:** Not applicable
**Recommendation:** 6

**Clarity, Quality, Novelty And Reproducibility:**

Clarity
- (f) A comprehensive review of BoBW under various settings (bandits versus MDPs and tabular versus function approximation).

Novelty
- (g) A nice observation that leverages the property of linear approximation to convert the problem to an online linear optimization problem.


**Strength And Weaknesses:**

- (a) The submission develops a new component (Algorithm 4 and Theorem 1) for the main algorithm.
- (b) What prevents the authors from directly analyzing the regret under the general function approximation setting?
- (c) The submission did a great job explaining the roles and jobs of the components of the main algorithm. The proof sketch is also easy to follow. It would be better if there were discussions about why other works fail, and how this work succeeds would better justify the technical contributions of this submission.
- (d) What prevents the authors from removing the known transition assumption? Why can't we leverage existing works on unknown transitions?
- (e) The paragraph above Section 5.1 discusses a way to handle the case when $\Pi$ is infinitely large (which is why function approximation was developed to replace the tabular methods). However, simply combining $Reg(K)$ with (14) and (15), the $|\pi|$ term is still there in the bound. Is there any misunderstanding?


**Summary Of The Paper:**

The submission studies the BoBW problem under a new setting. The submission proposes and analyzes the first BoBW algorithm for linear MDP with high-probability regret bounds (Algorithm 1). A secondary result is the first algorithm achieving a high-probability regret for the adversarial liner MDP (Algorithm 4).

**Summary Of The Review:**

The submission has substantial contributions to the BoBW problem. However, if allowing $|\Pi|$ to be infinity is the distinctive advantage of function approximation, the concern (e) raised above should be resolved. Therefore, I would like to recommend weak acceptance in the current stage.

---

> ### Author Response · Authors · 2022-11-18
> **Reply to Reviewer WXEg**
>
> We thank the reviewer for the valuable suggestions and comments. Please find our reply below.
>
> -general function approximation
>
> We agree that a general function approximation setting would be more desirable since it has larger approximation power. But to the best of our knowledge, the problem formulation as well as the theoretical guarantee for general function approximation is open even in the separate adversarial setting. To build a better theoretical understanding of the BoBW algorithm, we think it would be natural and also significant to start from linear function approximation, as many approaches for general function approximation are also inspired by that for linear function approximation (Wang et al., 2021; Jin et al., 2021a). It is worth noting that deriving BoBW results in a linear MDP setting is also a non-trivial problem given previous results in tabular cases, e.g., the BoBW algorithm for tabular bandits has been derived from 2012 but remains open for linear bandits until 2021.
>
> -discussions about why other works fail, and the technical contributions
>
> Thanks for your comment. We added this discussion in Section 5 in the revision.  And you can also refer to the reply to Reviewer GWBz for the corresponding discussion.
>
> -challenge in unknown transition setting
>
> It is worth noting that the linear MDP problem in an unknown transition case is still not well solved even in the separate adversarial setting. The state-of-the-art regret guarantee for this setting is $O(T^{14/15})$ achieved by Luo et al. (2021). The main hardness comes from that the agent needs to simultaneously learn the unknown transition and adapt to adversarial rewards. And whether a better regret guarantee can be derived is still an open problem. Since the BoBW problem is much harder than optimizing in a separate setting as the agent not only needs to learn the separate setting well but also adapt to different environment types, we first consider the known transition case.
> Our current algorithmic design highly depends on the observation that the value of a policy $\pi$ can be written as the inner product of the known visitation feature and an unknown parameter $\theta$. In the unknown transition case, if the agent has access to a simulator (given $\pi$, observe one trajectory by following $\pi$), it can still estimate the visitation feature with enough times of calling simulators and then run the algorithm with the estimated features. But if the simulator is unavailable, such an approach may fail and we conjecture a new algorithmic design is required for this case.

---

### Official Review · Reviewer_GWBz · 2022-11-01

**Confidence:** 3
**Correctness:** 4
**Technical Novelty And Significance:** 2
**Empirical Novelty And Significance:** Not applicable
**Recommendation:** 3

**Clarity, Quality, Novelty And Reproducibility:**

This work is original, extending previous works on best-of-both-worlds results for bandits and tabular MDP to the setting of linear MDP. For novelty, the authors should discuss the unique challenges in the setting of linear MDP and the corresponding solutions. For clarity and quality, I suggest revising the algorithm section for better readability. Also, the authors should explicitly and clearly specify the assumptions for each lemma and theorem.

**Strength And Weaknesses:**

Strength:

- The authors have done a good job reviewing the related literature.
- This paper studies an interesting problem of simultaneously learning stochastic and adversarial linear MDP, and the result presents a best-of-both-worlds guarantee.
- The theoretical analysis provides a high-probability regret guarantee for adversarial linear MDP.

Weaknesses:

- Given the existing literature on RL with function approximation, I believe the setting of linear MDP is already well-motivated. So for the introduction, the authors should discuss more the motivation of best-of-both-world type algorithm and, more importantly, explain the challenges in the algorithm design and theoretical analysis, compared with existing results.
- Following the previous point, please explain the technical challenges induced by the setting of linear MDP and the novelty of the proposed method.
- The description of the proposed algorithm in Section 4 is a bit difficult to follow. I would suggest the authors put more effort into reorganizing this section for readability. More specifically, Algorithm 4 seems to be an important subroutine, but it is not clearly explained what this algorithm is doing and why we need it. The description on page 5 could be more carefully structured using some highlighted words/sentences. Also, please provide references to existing related methods in other settings and compare them.
- It seems crucial to assume a finite policy set. Why is this a reasonable assumption for linear MDP? For example, consider the LSVI-UCB algorithm where the policy is given by the greedy policy w.r.t. the estimated Q-function, and in this case, the policy set is infinite. Even if using a covering argument for the policy set, it would introduce additional dependence on the dimension. Please justify this assumption and discuss its limitation in detail.

**Summary Of The Paper:**

This paper studies linear MDP with possibly adversarial rewards. The authors propose a detection-based algorithm that can simultaneously learn stochastic and adversarial linear MDP. Assuming a known probability transition, it is shown that the proposed algorithm can achieve logarithmic regret in the stochastic case and sublinear regret in the adversarial case.

**Summary Of The Review:**

I think this paper studies a meaningful setting and the results are solid. But the overall writing can be further improved, and see detailed comments above. I didn't carefully check the proof in the appendix.

Besides, I have a few more questions as follows:

- In the definition of the Q-function, normally people would include the reward received at the current step, i.e., $r_{k,h}(s,a)$
- The loops in Algorithm 1 and 2 are a bit weird. Should provide a stopping criterion, otherwise it's not clear when the algorithm ends.
- If assuming a known transition, do we still need to require the transition probability to be linear? I think without the linear assumption on the probability transition, the value function can still be written as a summation of inner products between the expected feature vectors and the parameters of the rewards.



---
Other minor problems:

- Near the bottom of page 5, should be 'standard least square estimators'
- Under Algorithm 3, should be 'fool the algorithm'

---

> ### Author Response · Authors · 2022-11-18
> **Reply to Reviewer GWBz**
>
> We thank the reviewer for the valuable suggestions and comments. Please find our reply below.
>
> -writing: the motivation of best-of-both-world type algorithm
>
> We thank the reviewer for the suggestion. We mainly discuss BoBW problem in Section 2 and we added a discussion on the challenge and novelty in Section 5 in the revision. It would be a different story by focusing on the BoBW problem in the introduction and we will modify it in the next version.
>
> -technical challenges and novelty
>
> There are mainly two types of algorithms to deal with the BoBW problem: the switch-based method which actively detects the environment type, e.g., Bubeck and Slivkins (2012), and the FTRL-based method which adapts to different environments, e.g. Zimmert et al. (2019). The approach in Bubeck and Slivkins (2012) first assumes the setting to be stochastic and would detect whether a policy's value has changed. Such an approach in our setting brings an $O(\sqrt{\Pi})$ dependence in the regret for adversarial setting and is not idealistic as the policy set size can be large. And the success of FTRL for BoBW mainly relies on a self-bounding inequality that bounds the regret by the chosen probabilities of policies. But such a technique is challenging with linear structure. As discussed by Lee et al. (2021), even for the single-state linear bandit setting, connecting FTRL with OP is hard.
> Our provided approach relies on a novel observation that the value of a policy can be written as the inner product of the visitation feature and an unknown reward parameter. From this view, we are able to reduce the problem to linear optimization and existing techniques for linear optimization can be used. It is worth noting that reducing a hard problem to a well-understood setting is also a typical way to solve the problem. For example, the widely studied occupancy measure-based approach also reduces harder MDP to the linear optimization problem and aims to optimize the policy by optimizing the occupancy measure. We have to emphasize that we are the first to introduce this type of linear optimization for the regret minimization problem in MDP and we believe such reduction may be of independent interest.
>
> -finite policy set, and additional dependence brought by covering
>
> When an appropriate covering is established, the log size of the covering set would be of order $O(dH)$ with the help of linear structure, e.g., the construction proposed by Reviewer PB4v. Thus though bringing additional dependence on $d,H$, it does not affect the regret dependence on $T$ and also does not influence our BoBW guarantee.
> We agree that removing the finite policy set assumption would be more general. Our approach is the first attempt to derive BoBW results for linear MDP with small policy classes. Though not the most general, the linear structure still improves the efficiency by improving the dependence on $SA$ in tabular cases to $d$. And when studying a harder problem, it is also natural to first start from a simpler case and find some intuition behind such cases. For example, literature on decentralized Markov games also starts from a small policy class.
>
> Learning Markov Games with Adversarial Opponents: Efficient Algorithms and Fundamental Limitshttps://arxiv.org/pdf/2203.06803.pdf
> Decentralized Optimistic Hyperpolicy Mirror Descent: Provably No-Regret Learning in Markov Gameshttps://arxiv.org/abs/2206.01588
>
> -the linear structure of known transition
>
> Thanks for your comment. Yes, the transition need not be linear since we only use the known visitation feature. We modified it in the revision.
>
> -writing: definition of Q
>
> Thanks for your comment. We modified the definition of Q and V by taking the sum from the current step $h$.
>
> -writing: loops in alg1,2
>
> Thanks for your comment. The episode index is common in Algorithms 1 and 2. Since the total horizon $K$ can be unknown, we do not include it as a stopping condition in algorithm charts. Each time Algorithm 2 returns, Algorithm 1 would end up the current while loop and enter in the next loop. If the total horizon is reached, the whole algorithm is stopped.
>
> -writing: the assumption of each lemma, theorem
>
> Thanks for your suggestion. Since all assumptions are formulated in Section 3 (Setting), we do not introduce them again in each lemma and theorem.
>
> -other typos
>
> Thanks for pointing out the typos. We modified them in the revision.

---

> > ### Comment · Reviewer_WXEg · 2022-11-20
> > **A thought on finite-sized policy space**
> >
> > Thank you very much for the feedback. Two other reviewers also mentioned the assumption on the policy space. From the review comments and authors' feedback,  a method for finite-sized policy space still does not answer the long-standing open problem of answering the potential of linear function approximation. For BoBW with finite-sized policy space, means other than linear function approximation might solve BoBW well too. When studying its potential, I believe we should keep the essence of linear function approximation -- a method handling a compact space.

---

> > ### Comment · Reviewer_GWBz · 2022-11-20
> > **Thank you for the response!**
> >
> > First I would like to thank the authors for the clarifications.
> >
> > However, I respectfully disagree that it should be regarded as 'a novel observation' that the value of a policy can be written as the inner product of the visitation feature and an unknown reward parameter. In my opinion, this directly follows from the assumption of known transition and sidesteps the critical question of estimating the transition, which is arguably one of the most important problems in RL, so this is even stronger than what's required for the occupancy measure-based approach where we need to estimate the occupancy measure.
> >
> > Indeed, by decomposing the value function in this way, the MDP problem is reduced to a linear bandit problem, where we view the stack of the visitation feature $\phi_\pi = (\phi_{\pi,1}, \ldots, \phi_{\pi,H})$ as the context vector (which is available because the transition is known) and the stack of reward parameters $\theta_k=(\theta_{k,h}, \ldots, \theta_{k,H})$ as the reward parameter. Now all the possible $\phi_\pi$ can be seen as the decision set induced by the policy set. From this point of view, I feel that the linear MDP problem has been largely simplified.

---

### Author Response · Authors · 2022-11-23
**Reply to all reviewers**

We thank all reviewers for their responses. We agree that deriving a BoBW algorithm under general policy space would be more desirable. Since the state-of-the-art guarantee for linear MDP in the adversarial setting with unknown transition is O(T^{14/15}), which may be still not well theoretically understood, we first start from the known transition case. In such a case, decomposing the value function into the inner product of the transition-visitation feature and unknown reward parameter is a good way to solve the problem. And our approach is the first such attempt to simultaneously learn in stochastic and adversarial settings. Deriving an FTRL-type algorithm may help for general policy space. However, this problem is even open in linear bandits where no state transition happens. We would also discuss the challenges more in a future revision.

---

### Decision · Program_Chairs · 2023-01-20

**Decision:**

Reject

**Justification For Why Not Higher Score:**

The reviewers found that the finite policy set assumption is too strong.

**Justification For Why Not Lower Score:**

N/A

**Metareview: Summary, Strengths And Weaknesses:**

This paper proposes an algorithm for linear MDP with the known transition, which can simultaneously achieve O(polylog⁡K) regret in the stochastic reward setting and O(\sqrt{K}) regret in the adversarial reward setting.

Strengths:

+ The first RL algorithm with linear function approximation that works for both stochastic reward and adversarial reward under known transition.

Weaknesses:

- The finite policy set assumption is very strong.

- Under the known transition assumption, the contributions of this work given prior work in BoBW linear bandits seems incremental.

Even after author response, this paper does not gather sufficient support from the reviewers. Thus I recommend rejection.

**Summary Of Ac-Reviewer Meeting:**

N/A